# Pathogenetic Features and Current Management of Glioblastoma

**DOI:** 10.3390/cancers13040856

**Published:** 2021-02-18

**Authors:** Hong-My Nguyen, Kirsten Guz-Montgomery, Devin B. Lowe, Dipongkor Saha

**Affiliations:** Health Sciences Center, Department of Immunotherapeutics and Biotechnology, Jerry H. Hodge School of Pharmacy, Texas Tech University, Abilene, TX 79601, USA; My.Nguyen@ttuhsc.edu (H.-M.N.); kguzmont@gmail.com (K.G.-M.); devin.lowe@ttuhsc.edu (D.B.L.)

**Keywords:** glioblastoma, GBM pathogenesis, heterogeneity, targeted therapy, immunotherapy

## Abstract

**Simple Summary:**

Glioblastoma (GBM) is the most common form of primary malignant brain tumor with a devastatingly poor prognosis. Tumor heterogeneity (cellular, molecular and immune) is the major obstacle to current treatment failure. We revisited the recent literature to understand the heterogeneous features of GBM and their potential role in treatment resistance. This review provides a comprehensive overview covering the GBM’s pathogenetic features, currently available treatment options and the treatments currently under development in the clinic.

**Abstract:**

Glioblastoma (GBM) is the most common form of primary malignant brain tumor with a devastatingly poor prognosis. The disease does not discriminate, affecting adults and children of both sexes, and has an average overall survival of 12–15 months, despite advances in diagnosis and rigorous treatment with chemotherapy, radiation therapy, and surgical resection. In addition, most survivors will eventually experience tumor recurrence that only imparts survival of a few months. GBM is highly heterogenous, invasive, vascularized, and almost always inaccessible for treatment. Based on all these outstanding obstacles, there have been tremendous efforts to develop alternative treatment options that allow for more efficient targeting of the tumor including small molecule drugs and immunotherapies. A number of other strategies in development include therapies based on nanoparticles, light, extracellular vesicles, and micro-RNA, and vessel co-option. Advances in these potential approaches shed a promising outlook on the future of GBM treatment. In this review, we briefly discuss the current understanding of adult GBM’s pathogenetic features that promote treatment resistance. We also outline novel and promising targeted agents currently under development for GBM patients during the last few years with their current clinical status.

## 1. Introduction

Glioblastoma (GBM) is one of the most common forms of primary malignant brain tumor and has a very poor prognosis with an average patient survival lasting only 12–15 months [1,2]. This bleak outlook is due in part to the challenges that are presented by the anatomical location of the tumor as well as the heterogeneity of GBM cells and their rapid growth rate [3,4]. Although GBM is known to affect both adults and children, the incidence of GBM increases with age peaking in the 1970s [5]. Cancer incidence is roughly 2-3 individuals per 100,000 cases each year in the United States, with rates increasing slightly based on patient age [6,7,8]. There is also a slightly higher rate of incidence in men versus women, with men being 1.6 times more likely to develop GBM [9]. GBM accounts for approximately 46% of all diagnosed brain tumors and causes around 2.7% of all cancer-related deaths [3]. In fact, it is ranked as the third most common cause of death from cancer in patients between 15 and 34 years [7]. There are currently four grades of gliomas classified by the World Health Organization (grades I-IV) [7,10]. Grade IV gliomas are the most aggressive and invasive forms and are responsible for the poorest prognoses [10,11]. GBM typically refers to these grade IV gliomas and can be subdivided into primary and secondary types [12,13]. Although primary and secondary gliomas share similar histological characteristics, they have very different genetic profiles [14]. Primary GBM constitutes approximately 90% of GBM cases and is considered a de novo pathway of multistep tumorigenesis from glial cells while secondary GBM develops from lower-grade and pre-existing tumors such as diffuse astrocytomas [15]. Of the two, primary GBM is generally found to be more malignant than secondary GBM [16], and men are somewhat more likely to present with primary GBM while women are more likely to be diagnosed with secondary GBM [17].

The standard treatment options for GBM include surgery, chemotherapy, and radiation. However, even with these interventions, GBM still carries a dismal prognosis [18,19]. Diverse pathogenetic features and immunosuppression are two major contributors of current treatment failure. Although many studies have attempted to design effective treatments around these challenges, none have been developed that are capable of achieving long-term patient survival without causing unwanted damage to the delicate cells and neuronal tissues of the brain [4]. Over the past several years, targeted therapies and immunotherapies have shown great achievement in GBM management with promising results in clinical trials [18,19,20,21]. Other therapies in development include nanotechnology-based innovations, photodynamic strategies, gene therapy, and local destruction of the tumor via genetically modified bacteria or controlled hyperthermia. In this review, we discuss the current understanding of GBM’s pathogenetic features (i.e., cellular, molecular, and immunosuppressive properties) that contribute to treatment resistance. We also outline novel targeted therapies, different immunotherapeutic approaches, and a number of other promising/emerging treatment strategies for adult GBM that are currently under development.

## 2. Pathogenetic Features

### 2.1. Cellular Heterogeneity, Tumor Vascularity, and Extracellular Matrix

GBM is highly heterogeneous, both intrinsically and intratumorally, with multiple factors driving its development and growth [22,23]. The tumor grows in an infiltrative manner, making it difficult to distinguish and surgically remove from the normal brain [24]. Over half of primary GBMs occur in the four lobes of the brain, with smaller parts also found in the brain stem and spinal cord [7,16,25]. Among those occurring in the brain, approximately 25% are located in the frontal lobe, 20% in the temporal lobe, 13% in the parietal lobe, and 3% in the occipital lobe [25]. However, while primary GBM can develop in any number of locations in the brain, secondary GBM is found primarily in the frontal lobe [12]. GBMs were once thought to be derived from neural stem cells (NSCs), NSC-derived astrocytes, and oligodendrocyte precursor cells (OPCs) [26]. However, new evidence suggests that astrocyte-like NSCs in the subventricular zone (SVZ) are the cell of origin for GBM [27,28,29,30]. Lee and colleagues established that astrocyte-like NSCs harboring low mutational levels can migrate from the SVZ to a distinct site and develop high-grade gliomas [29]. In addition, NSCs in the SVZ were found to carry limited self-renewal abilities, thus, being able to escape replicative senescence and acquire driver mutations over time that play a role in GBM recurrence [29]. The cells that make up GBM are small, polymorphic, and anaplastic. In appearance, they are usually polygonal or spindle-shaped, with indistinct borders and acidophilic cytoplasm [28,31]. These cells are known to have clumped chromatin with distinct nucleoli, oval or elongated nuclei of varying sizes, and an increased nuclear to cytoplasmic ratio. In addition, some cells may be binuclear or multinucleated and may have large lipomatous vacuoles [28,31]. The endothelial cells of GBM are also unique in that they overlap focally and are heterogeneous in appearance. While normal brain endothelial cells lack Weibel–Palade bodies, which are the storage granules of endothelial cells, GBM endothelial cells (GECs) in new vasculature contain many of these granules [31]. Weibel–Palade bodies in GECs secrete von Willebrand Factor (VWF)—a pro-angiogenic factor that is associated with a three-fold higher risk of death in GBM patients [32]. GBM also contains a sub-population of cancer cells that display stem-cell qualities designated GBM stem-like cells (GSCs) [27,33,34]. Differentiation of these GSCs leads to an incredibly diverse population of cell types such as microglia-like cells [35,36,37,38,39] and oligodendrocyte-like cells within the tumor [40,41]. GSCs possess a number of other characteristics including self-renewal, increased proliferation and migration, suppression of immune responses, support of angiogenesis, and increased radio- and chemoresistance [27,33,34]. Because of these abilities and characteristics, GBMs often recur from GSCs and are often distinctly unique from the original glioma with a developed resistance to previously applied treatments [33].

GBM is a highly vascularized tumor [42] that occurs via multiple mechanisms such as vessel co-option, angiogenesis, vasculogenesis, endothelial cell trans-differentiation, and vascular mimicry [43]. Vessel co-option is a non-oncogenic process where tumor cells use nutrients from the pre-existing vasculature of normal/healthy tissues to support tumor growth and tumor cell survival [44]. This process involves the movement of tumor cells along the vasculature and contributes greatly to the infiltrative growth of gliomas [45,46]. Some pre-clinical studies show that vessel co-option is a preferred mechanism of vascularization in early-stage gliomas, which later may or may not switch to the process of angiogenesis (i.e., development of new blood vessels) in GBM [47,48,49]. Angiogenesis in GBM can be induced by different factors such as basic fibroblast growth factor (bFGF), platelet-derived growth factor (PDGF), transforming growth factor-beta (TGF-β), and angiopoietins [50]. However, tumor angiogenesis is oftentimes nonproductive due to abnormal vessels that lead to vascular occlusion [51,52] that causes a hypoxic environment surrounding the failed vessel and ensuring central necrotic tissue [31]. Tumor cells then actively migrate away from this central hypoxia, resulting in the formation of a zone of hypercellular tissue surrounding an area of necrosis (defined as a pseudopalisade). Pseudopalisades secrete hypoxia-inducible factor (HIF-1), vascular endothelial growth factor (VEGF), hepatocyte growth factor (HGF), matrix metalloproteases (MMPs), and interleukin-8 (IL-8) that promote microvascular proliferation, angiogenesis, and tumor expansion [51]. Clinically, pseudopalisades are thought to be a major contributor to the rapid disease progression and poor survival rate and response to treatment [53]. GBM endothelial cells can be derived from bone marrow (BM)-derived endothelial progenitor cells (EPCs) or GSCs through vasculogenesis [54] and endothelial cell trans-differentiation, respectively [37,55]. GSCs also possess an ability to form a non-endothelial tube-like structure that mimics the tumor vasculature and supports tumor growth—a process otherwise known as vascular mimicry that is associated with poor prognosis in GBM patients [56,57]. Importantly, the steps of GSC-induced tumor vascularization can also be potentiated by GBM standard-of-care interventions [58,59]. For instance, traditional temozolomide (TMZ) chemotherapy can promote neovascularization by inducing chemotherapeutic stress, which helps transdifferentiate GSCs into endothelial cells and vascular mimicry [58]. Similar to TMZ, ionizing radiation also promotes GSCs to transdifferentiate themselves into endothelial cells through the Tie2 signaling pathway [59]. GSCs may also survive or be enriched by anti-angiogenic therapy, leading to tumor recurrence and anti-angiogenic therapy (AAT)-resistant tumors [60,61].

The relative volume of extracellular space in the normal brain is about 24%, which can be increased by up to 48% during the progression from lower to higher-grade gliomas [62]. The expansion of this extracellular space is thought to create an ideal environment for tumor migration and invasion [62]. GBM cells produce different ECM components such as hyaluronic acid (HA), matrix metalloproteinase-2 (MMP2), matrix metalloproteinase-9 (MMP9), integrins, tenascin-C [63], and fibronectin [64] to degrade and remodel the ECM for tumor invasion [65,66,67]. Adhesion of tumor cells to ECM proteins is regulated by transmembrane receptors known as integrins like αvβ3 and αvβ5. Both receptors are highly expressed on GBM cells and GBM endothelial cells [68], and their overexpression is associated with poor overall survival (OS) in GBM patients solely treated with standard chemotherapy [69]. Inhibiting integrins blocks angiogenesis, tumor invasion, stemness, and immunosuppression [68]. In addition, integrins such as α5β1 are overexpressed in GBM tumors that have reduced p53 activity due to TMZ treatment. The negative cross-talk between α5β1 and p53 attributes to TMZ resistance [70]. GBM cells communicate with surrounding tumor and non-tumor cells by secreting special structures known as extracellular vesicles (EVs) that are often encapsulated with different factors to promote tumor growth [71,72]. For example, EVs derived from GBM cells stimulate normal astrocytes to be converted into a tumor-supportive phenotype via p53 and MYC signaling pathways, leading to ECM destruction [71]. GSCs secrete different pro-angiogenic factors such as VEGF or miRNAs such as miR-21 or miR-26a in EVs to promote angiogenesis [73,74,75] and facilitate tumor growth under hypoxic conditions [76]. The mRNA and protein expression of numerous invasion-associated ECM molecules such as Fms-related tyrosine kinase 4 (FLT4), mouse double minute 2 homolog (MDM2), and MMP-2 vary between different GBM subtypes, and such ECM compositions can be used as prognostic indicators for patients with GBM [77].

### 2.2. Molecular Heterogeneity

The Cancer Genome Atlas (TCGA) Project created a GBM classification system based on 600 genes that were sequenced from 200 human tumor samples [78]. This system has shed new light on the complexity of the genetic profile of GBM and led to the discovery that molecular alterations can be used to distinguish primary and secondary gliomas. Primary and secondary GBM are histologically similar but genetically and epigenetically different. Primary GBMs, which are also designated as isocitrate dehydrogenase wild-type GBMs (IDH-WT), are characterized by (i) overexpression of epidermal growth factor receptor (EGFR), (ii) mutated telomerase reverse transcriptase promoter, p53, and phosphate and tensin homologue (PTEN) genes, and (iii) loss of chromosome 10q and cyclin-dependent kinase inhibitor 2A (CDKN2A) gene [11,79]. Secondary GBMs (also referred to as IDH-mutant GBMs) typically harbor mutations in p53 and isocitrate dehydrogenase 1 (IDH1) genes alongside the loss of chromosome 19q [79]. IDH-WT GBMs are more common, respond poorly to treatment, and have an overall lower survival rate than IDH-mutant GBMs [79,80].

GBM can also be categorized into four tumor subtypes: proneural, mesenchymal, classical, and neural [81]. The signature gene alternations for each subtype are listed in Table 1. Briefly, proneural GBM is characterized by the alteration of platelet-derived growth factor receptor alpha (PDGFRA) and mutated IDH1 along with higher expression of certain proneural markers (e.g., SRY-related HMG-box genes [SOX], doublecortin [DCX], delta-like canonical Notch ligand 3 [DLL3], achaete-scute family BHLH transcription factor 1 [ASCL1], transcription factor 4 [TCF4] and oligodendrocytic development genes (e.g., PDGFRA, NK2 homeobox 2 [NKX2-2], oligodendrocyte transcription factor 2 [OLIG2] [81,82,83]. Mesenchymal GBM is featured by focal hemizygous deletions of a chromosomal region at 17q11.2, co-mutations in neurofibromin 1 (NF1) and PTEN genes, and enrichment of tumor necrosis factor (TNF) super family pathway and nuclear factor kappa B (NF-κB) pathway genes (e.g., TNFRSF1A associated via death domain [TRADD], v-rel avian reticuloendotheliosis viral oncogene homolog B [RELB], TNF receptor superfamily member 1A [TNFRSF1A]) [81]. Mesenchymal GBM is also linked to higher expression of mesenchymal and astrocytic markers such as CD44 or c-mer proto-oncogene tyrosine kinase (MERTK), which promote epithelial–mesenchymal (EMT) transition [81,84]. Classical GBM is characterized by amplification of chromosome 7 and loss of chromosome 10 [81]. Overexpression of EGFR, absence of TP53 mutations and focal 9p21.3 homozygous deletion that targets cyclin-dependent kinase inhibitor 2A (CDKN2A) are also predominant features in classical GBM [81]. The common markers for classical GBM include the neural precursor and stem cell markers nestin, notch (neurogen locus notch homolog protein 3 [NOTCH3], jagged canonical notch ligand 1 [JAG1], O-Fucosylpeptide 3-Beta-N-Acetylglucosaminyltransferase [LFNG] and sonic hedgehog (smoothened, frizzled class receptor [SMO], growth arrest-specific 1 [GAS1], GLI family zinc finger 2 [GLI2])) [81]. Lastly, the major properties of neural GBM include neuron projection, axon and synaptic transmission, and expression of neuronal markers such as neurofilament light (NEFL), gamma-aminobutyric acid type A receptor subunit alpha1 (GABRA1), synaptotagmin 1 (SYT1), and solute carrier family 12 member 5 (SLC12A5) [81].

The distinction of the four GBM subtypes by the TCGA also revealed common themes in altered-signaling pathways that dictate cell growth and regulation, DNA repair, and apoptosis (Figure 1) [78,79,80]. In a TCGA-based study, GBM specimens routinely contained aberrations in the following signaling pathways: p53 (87%), RB (78%), and RTK/Ras/PI3K (88%) [78,79,85]. Disruptions in p53 signaling are associated with increased cell migration, invasion, and survival [86]. The Rb pathway is controlled by phosphorylation of Rb by cyclin D, cyclin-dependent kinase 4 (CDK4), or CDK6 [87]. CDK4 and CDK6 are further regulated by CDK inhibitors (CDKN2A, CDKN2B, CDKN2C), and mutations to CDKN2A or Rb1 cause cell cycle disruption, allowing uncontrolled cell proliferation and apoptosis evasion [87]. RTKs (once activated) regulate cell proliferation, differentiation, angiogenesis, and survival through either downstream PI3K/AKT/mTOR or Ras/MAPK/ERK signaling [88]. Mutations or alterations in genes that encode RTKs (e.g., epidermal growth factor receptor [EGFR], vascular endothelial growth factor [VEGF], or insulin-like growth factor 1 receptor [IGF-1R]), lead to unregulated cell proliferation and survival [80,88].

### 2.3. Immunosuppressive Features

The CNS is separated from the circulatory system by the blood–brain barrier (BBB) that effectively prevents the passive diffusion of molecules. The CNS was previously viewed as an immune-privileged site [89]. However, recent discoveries have identified functional lymphatic vessels in the brain’s dura matter that drain cerebrospinal fluid into cervical lymph nodes [90,91]. In this way, tumor-derived antigens can concentrate within lymph nodes to stimulate immune cell responses involving T cells [90]. However, after infiltrating into the GBM tumor, T cells become dysfunctional via different mechanisms that could include senescence, tolerance, anergy, exhaustion, or ignorance that ultimately leads to poor GBM prognosis (Figure 1) [92]. T cell senescence is associated with the loss of the co-stimulatory molecule CD28, which could occur as a result of telomere damage, regulatory T cell (Treg) interaction, or metabolic competition in the tumor microenvironment (TME) [93,94]. T cell tolerance is mediated by Fas ligand (FasL)-induced apoptosis, recruitment of Tregs, and upregulation of factors that dampen T cell effector function such as cytotoxic T-lymphocyte antigen-4 (CTLA-4), indoleamine 2,3-dioxygenase 1 (IDO-1), and signal transducer and activator of transcription 3 (STAT3) [92,95,96,97]. Anergy is mainly observed in infiltrating CD4^+^ T cells and is due in part to RAS/MAPK dysfunction or inefficient Zap70 kinase activity that impairs interleukin-2 (IL-2) production (and, hence, T cell activation/proliferation) [92]. The co-expression of immune checkpoint molecules such as programmed cell death 1 (PD-1), lymphocyte-activation gene 3 (LAG3), and T-cell immunoglobulin mucin-3 (TIM-3) are also important contributors to T cell exhaustion in GBM [92]. For example, GBM may express PD-L1 and induce T cell exhaustion through interaction with its cognate receptor PD-1 expressed by T cells [98,99]. T cell ignorance as a result of sphingsosine-1-phosphate receptor 1 (S1P1) loss that prevents T cells from trafficking to the tumor site, disrupting T cell-mediated anti-tumor immunity and contributing to tumor progression [92]. Additionally, T cell function can be further impaired through the GBM vasculature instituting physically-induced constraints (e.g., hypoxia) and promoting infiltration of immunosuppressive immune cells (such as macrophages, neutrophils, and myeloid-derived suppressor cells [MDSCs]) [100]. Such immune cell subsets release pro-inflammatory mediators and cytotoxic cytokines, growth factors, bioactive lipids, hydrolytic enzymes, matrix metalloproteinases, reactive oxygen intermediates, and nitric oxide to support the continued growth of cancer cells [100]. GBM cells also release a number of immunosuppressive factors, including TGF-β, prostaglandin E (PGE), interleukin-1 (IL-1), interleukin-10 (IL-10), and fibrinogen-like protein 2 (FGL2) that suppress DC priming/activation of immune effector cells [101]. Tumor-derived immunosuppressive factors also help recruit pro-tumoral M2 macrophages and Tregs, which further secrete TGF-β1 and IL-10 and eventually suppress T cell effector functions [102,103,104,105]. In a recent study, an extensive level of tumor immunosuppression in GBM (excluding grade II and III gliomas), is found due to the presence of blood-derived macrophages, tumoral expression of programmed cell death ligand 1 (PD-L1), and T cell expression of PD-1 [106]. The same study reported that bone marrow-derived macrophages migrate to the tumor site and accumulate centrally in GBM lesions, exerting strong immune suppression by releasing iron that is necessary to maintain tumor cell survival and tumor progression. In contrast, resident microglia exert little to no immunosuppressive function [106].

## 3. Current Treatment

### 3.1. Standard of Care and Other FDA Approved Treatments

The current standards of care for GBM include maximal resection surgery, radiation, and temozolomide (TMZ) therapy—with TMZ and radiation being commenced within 30 days post-surgery [107,108,109]. Unfortunately, GBM response to TMZ varies between patients, and many types of GBM carry resistance to the compound. Treatment resistance to standard therapies is likely due to a combination of upregulated DNA repair mechanisms and the presence of GSCs that maintain an ability to self-renew and differentiate [110]. TMZ resistance also appears to be driven by the DNA repair enzyme O6-methylguanine-DNA methyltransferase (MGMT) that repairs DNA alkylation since patients bearing MGMT genes with methylated promoters seem to be more responsive to TMZ treatment [110]. However, TMZ damages both tumor and normal cells and does not eliminate GBM, so, options for alternative treatments are desperately needed [4].

Other treatments approved by the FDA for use in GBM therapy include bevacizumab and tumor-treating fields (TTFs). Bevacizumab is a humanized monoclonal antibody (mAb) that targets the angiogenic factor VEGF and was the first anti-angiogenic drug approved for patient use after showing increased overall survival in colorectal and non-small-cell lung cancers when combined with chemotherapy [111,112]. In addition, the antibody was observed to be safe in patients and mediate effective anti-tumor responses in Phase II clinical trial for recurrent GBM when combined with the chemotherapeutic drug irinotecan [113]. However, in two Phase III trials conducted in newly diagnosed GBM patients, treatment with bevacizumab in addition to either radiation or radiation plus TMZ showed no significant difference in overall survival compared to the placebo [114,115]. Although progression-free survival was better with treatment, patients suffered a higher frequency of adverse events and poorer quality of life [115].

TTFs are a non-invasive and anti-mitotic FDA-approved strategy for newly diagnosed cases of GBM (i.e., as adjuvant therapy) or recurring disease [116,117]. It involves using alternating electrical fields with a frequency range of 100-300 kHz and an intensity of 1 to 3 V/cm to interfere with the functions of rapidly dividing cancer cells, causing cessation of cell division and ultimately leading to cell death [116,118]. The theory behind this treatment is that the electrical fields create space between the growing ends of microtubules and tubulin dimers, thus, interfering with microtubule polymerization of the mitotic spindle [117]. Recently, TTF has been tested in combination with current standard-of-care in newly diagnosed GBM patients. Concurrent administration of TTF/radiation/TMZ followed by adjuvant TMZ/TTF demonstrated safety and promising preliminary efficacy [119], which warrants further clinical investigation in a larger patient cohort. In fact, a clinical study is currently ongoing utilizing this triple combination in 60 newly diagnosed GBM patients (NCT03869242).

### 3.2. Hurdles with Current Treatments 

There are many aspects that make GBM difficult to effectively treat [120]. One complication is enabling the treatment drug to cross the BBB and reach the tumor [108]. It was previously thought that the BBB was uniformly disrupted in cases of GBM, and was, therefore, not an issue when designing treatment plans [108]. However, recent evidence suggests that a large portion of the BBB remains intact, presenting a challenge to many drug therapies [108]. Considering that drug molecules are unable to reach the tumor to potentiate effects, BBB transporters often remove most of the molecules that do manage to pass through [4].

The infiltrative and invasive growth of GBM also impedes complete surgical resection of tumor cells. Thus, secondary treatment is usually needed following surgery [121,122]. In addition, most GBMs that initially respond well to treatment recur after a period of a few months. Relapsed tumors generally have an even poorer overall survival and do not respond well to previously used treatments [123] as they acquire new mutations and evasive properties [124]. Tremendous efforts have been made to target those mutations by targeted therapies (discussed in more detail below).

Other major reasons for treatment failure can include: (i) GBM is extremely immunosuppressive [125,126,127], (ii) tumor cells contain a low somatic mutational load [78,128], which could explain poor responses to immune checkpoint blockade to the anti-PD-1 antibody (CheckMate-143), and (iii) presence of GSCs that help drive resistance to radiotherapy [129,130], chemotherapy [131], and anti-VEGF therapy [132]. Although TMZ is effective against MGMT-negative GSCs [133], the drug is incapable of eliminating MGMT-positive GSCs [134]. Resistance to TMZ in particular and other therapies mediated by GSCs also rely on an ability to regulate various miRNA molecules that can remodel different signaling pathways in response to treatment [135,136]. GSC plasticity also allows differentiation into a slow-cycling and persistent cellular state that can escape cytotoxicity from different targeted therapies [137]. Treatment-resistant GSCs further induce immunosuppression by recruiting M2-like tumor-associated macrophages (TAMs) and Tregs into the TME [138,139]. Lastly (iv), while effective anti-tumor immunity in GBM is profoundly inhibited, possibly by promoting subsets of dysfunctional T cells through various mechanisms [92], it is important to note that current standard therapies (including TMZ and high-dose corticosteroids) might worsen GBM’s immunosuppressive status [140,141,142]. Thus, there is a need to develop newer forms of immunotherapy that overcome immunosuppression and boost the host’s anti-tumor immune responses [121].

## 4. Treatments in Development

### 4.1. Targeted Therapies

Based on dysregulated signaling in GBM, targeted therapies are mainly categorized to ablate: the p53, RB, and receptor tyrosine kinase (RTK) signaling pathways (Figure 2). In general, intra- and inter-tumoral heterogeneity of mutated signaling pathways in GBM incite resistant mechanisms to monotherapeutic treatment with targeted agents. While combination therapies to target multiple pathways is one potential route to overcoming resistance, developing improved better strategies to impact each individual mutational alteration in GBM are gaining interest [121].

#### 4.1.1. Targeting the p53 Pathway

The P53 pathway (including CDKN2A, MDM2 and TP53) is dysregulated in 85% of GBM tumors [78], with prevalence varying between different GBM molecular subtypes (proneural 54%, mesenchymal 32%, neural 21%, and classical 0%) [81]. Current options to target the p53 pathway in GBM include inhibiting the MDM2/p53 complex, restoring wt-p53 conformation and gain of function (GOF), and degrading Mut-p53 [86].

Inhibiting MDM2/p53 interaction to reactivate the anti-tumor function of p53 offers a promising approach for patients, with many MDM2 inhibitors currently in development for GBM preclinically (such as RG7112, MI77301, CGM097, and MK8242) and clinically (such as RG-7388 and AMG-232). AMG-232 treatment, either as a monotherapy or in combination with other chemotherapies, markedly increases activation of p53 signaling in tumors, making the drug the most potent MDM2 inhibitor to date [143]. Clinical trials for MDM2 inhibitors are currently running in both newly diagnosed and recurrent GBM patients (Table 2).

Cancer-associated p53 mutation is often a single amino acid substitution that favors cancer cell survival and tumor progression [144]. Since mutant p53 is associated with malignant progression, chemoresistance, invasion, cancer maintenance, and metastasis, GOF p53 mutations represent promising targets for novel cancer therapy development [145]. PRIMA-1 and its structural analogue PRIMA-1^MET^ (APR-246) are the most extensively studied compounds that restore wt-p53 conformation and p53 function [86]. Both drugs induce p21 expression, cell cycle arrest and apoptosis, and inhibit GBM stemness and growth [146,147,148,149]. While PRIMA-1 and APR-246 have been successfully tested in different cancer types, there is no clinical trial of APR-246 reported so far in GBM. Other drugs in this class include PK11007, PK7088, PEITC, ZMC1, COTI-2, CP-31398, small peptides (ReACp53 and pCAPs), and RETRA [86,145].

Another p53 targeting approach involves degrading mutant p53 by inhibiting interactions with the heat shock proteins (Hsp) Hsp70 and Hsp90. Histone deacetylase 6 (HDAC6) is required to form the mutant p53-Hsp70/Hsp90 complex. Thus, histone deacetylase inhibitors (HDACi) (such as trichostatin A, CUDC-101, CUDC-907, and vorinostat) used either alone or in combination with other drugs can disrupt this mutant p53-Hsp complex, resulting in the degradation of mutant p53 in GBM [150,151,152,153]. Detailed mechanisms and clinical significance of HDACi in GBM have been previously discussed [154]. Current clinical studies are focused on exploring the synergistic combination of HDACi and chemotherapy and/or radiotherapy in GBM patients as listed in Table 2.

#### 4.1.2. Targeting the Rb Pathway

The genes that are mostly altered in the RB pathway in GBM include CDK4, CDK6, CCND2, CDKN2A/B, and Rb1 [78]. The tumor suppressor pRb plays a crucial role in cell cycle progression through the regulation of cyclin-dependent kinases (CDKs) at the G1/S phase. Although mutation of cycD1-CDK4/6-Rb1 occurs in approximately 80% of GBM cases and is one of the top three most altered pathways [78,155], the essential function of this pathway in normal healthy cells has limited its potential as a target for GBM treatment. CDK inhibitor such as palbociclib directly suppresses phosphorylated Rb1 and induces cell cycle arrest and apoptosis [156] and include drugs such as palbociclib, ribociclib or abemaciclib that are FDA approved for other cancer types and are currently being clinically evaluated in GBM (Table 2) [157,158]. Palbociclib alone has not been effective in patients with recurrent GBM [157], indicating combination with other therapies is likely required to improve therapeutic outcomes [156,159]. In fact, combined CDK4/6 inhibition with radiotherapy resulted in an improved survival advantage over monotherapies in a mouse model of GBM, which was associated with a significant increase in γH2AX (a marker for DNA damage) and cleaved poly (ADP-ribose) polymerase (PARP) (a measure for apoptosis) [156]. Combined CDK4/6 and mTOR inhibition has also resulted in disruption of GBM metabolic pathways, leading to significant induction of apoptosis compared to single treatments [159]. 

#### 4.1.3. Targeting the RTK Pathway

##### Epidermal Growth Factor Receptor (EGFR) Inhibitors

EGFR mutations occur at high frequencies (57.4%) in GBM [78] and indicate poor prognosis [160]. The bulk of EGFR mutations in GBM are largely classified as *EGFRvI* (N-terminal deletion), *vII* (deletion of exons 14–15), *vIII* (deletion of exons 2–7), *vIV* (deletion of exons 25–27), and *vV* (deletion of exons 25–28) [161]. Among these mutants, *vII/vIII* are oncogenic [161], and *vIII* is the most common variant in GBM [78,162]. Unfortunately, previous attempts targeting EGFR in GBM have not been successful and may be a result of the diverse molecular features of the molecule [163]. In addition, Kwatra and colleagues [164] outlined the major flaws of failed clinical trials with first- and second-generation EGFR tyrosine kinase inhibitors (EGFR-TKIs): (i) EGFR-TKI gefitinib blocks the cell-surface receptor but does not abrogate downstream signaling, promoting alternate tumor growth signaling pathways; (ii) Failed clinical trials included both wild-type and EGFR-activated GBM patients instead of patients with activated EGFR or EGFR*vIII*; and (iii) Most EGFR-TKIs (e.g., erlotinib, gefitinib, afatinib, and lapatinib) are poorly penetrant to the brain [165]. However, a third-generation EGFR-TKI, AZD9291, has recently been approved for the treatment of non-small cell lung cancer, irreversibly binds with high affinity to EGFR*vIII*, and is ~10 times more efficient than first-generation EGFR inhibitors in inhibiting tumor cell proliferation in an orthotopic GBM model [166]. These findings suggest AZD9291 efficiently crosses the BBB and could be a useful EGFR-TKI for GBM patients.

Overall, current clinical approaches to specifically targeting EGFR and EGFR*vIII* include EGFR-TKIs, unmodified anti-EGFR antibodies (e.g., cetuximab, panitumumab, and nimotuzumab), engineered anti-EGFR antibodies (e.g., conjugated with a radioactive isotope [125I mAb 425] [167], antibody–drug conjugate (e.g., depatuxizumab mafodotin) [168], bispecific format [bscEGFR*vIII*xCD3]), anti-EGFR*vIII* vaccines (CDX-110), EGFR*vIII*-specific CAR-T cells, and RNA-based formulations that have been described elsewhere [162]. In general, strategies directed against EGFR/EGFR*vIII* have not yet gained clinical benefits in a majority of patients and would likely benefit from a clearer understanding of the molecular dynamics of EGFR/EGFR*vIII* cancer cell signaling [162].

##### Phosphatidylinositol-3-Kinase/AKT/Mammalian Target of Rapamycin (PI3K/AKT/mTOR) Inhibitors

PI3K/AKT/mTOR is an important signaling pathway to regulate cell growth, motility, survival, metabolism, and angiogenesis [169,170]. Impairment in PI3K/AKT/mTOR pathway in GBM can be caused by different mechanisms such as loss/inactivation of PTEN, mutation/amplification of PIK3CA, and activation of RTKs or oncogenes upstream of PI3K [78,171]. Although more than 50 PI3K inhibitors have been tested in various types of cancer, a limited number of drugs have entered into clinical trials for GBM patients [172]. Development of resistance to PI3K inhibition, reduced BBB permeability, and poor safety profile of PI3K inhibitors are major contributors to a less than pronounced clinical translation [171].

AKT stands as a midpoint between upstream and downstream regulation of cell growth and survival signals—the main mechanism of resistance to chemo- and radiotherapy, making its inhibition an attractive target for GBM [173]. AKT inhibitors can be sub-divided into lipid-based phosphatidyl-inositol analogues, ATP-competitive inhibitors, and allosteric inhibitors [173]. However, the AKT pathway is complex and promiscuous and contains different AKT isozymes that differ in function and tissue distribution, making any promise of selective AKT drugs difficult to realize. For example, inhibiting AKTs as a single agent has shown marginal effects in a phase II trial (NCT00590954). Disrupting mTOR, which controls downstream targets of phosphorylated Akt that regulate protein synthesis, cancer cell survival, invasiveness and GSC maintenance, chemotherapy resistance, and angiogenesis, is another attractive approach for GBM [174]. In fact, combining mTOR inhibitors with other therapies such as CDK4/6 [159] or MDM2 blockade [175] has provided some promise in GBM treatment. Along with instigating direct anti-proliferative activities following mTOR inhibition, mTOR disruption offers another potential anti-tumor mechanism [176]. For instance, mTOR is activated in 39% of tumor-associated microglial cells (as tested in 42 human GBM tumor specimens) but is downregulated following treatment with an mTOR inhibitor, which suggests mTOR disbarment in GBM patients might reduce the frequency of pro-tumorigenic M2-type macrophages [176]. However, the direct role of mTOR inhibitors on the immune system is controversial, since mTOR blockade can lead to either immunosuppressive or immunomodulatory effects depending on the cell types and nature of stimuli involved [177,178]. Further research is obviously warranted to possibly exploit mTOR inhibition for the improvement of GBM immunity.

##### Hepatocyte Growth Factor Receptor (HGFR/c-MET) Inhibitors

Interaction of hepatocyte growth factor (HGF) to MET triggers several downstream signaling pathways and promotes carcinogenesis [179]. MET amplification is detected in 1.6–4% of GBM patients [78,155] and its expression is associated with poor prognosis [180,181]. MET targeted therapies can either be mAb or small molecule inhibitors. Despite a better understanding of MET signaling and its associated resistance mechanisms, the clinical benefit of anti-MET therapies remains minimal. Results from a recent randomized, double-blinded, placebo-controlled, multicentered phase II study in patients with recurrent GBM showed that the anti-MET mAb onartuzumab failed to provide additional clinical benefit when added alongside bevacizumab [182].

##### Fibroblast Growth Factor Receptor (FGFR) Inhibitors

The FGF-FGFR signaling pathway regulates many biological functions, including cell proliferation, survival, and cytoskeletal regulation [183]. FGFR mutations are found in 3% of GBM cases and associated with poor overall survival in GBM patients treated with chemoradiation [184]. The heterogeneous expression of different FGFRs (type 1-4) in GBM, along with a lack of understanding of FGFR’s contribution to GBM progression make this target’s outlook for GBM less clear. Interestingly, recent research finds that GBM upregulates the FGFR pathway in response to dual blockade of MET and EGFR [185], suggesting that a more comprehensive drug cocktail (i.e., inhibiting FGFR, MET, and EGFR) might provide patients added benefit.

##### Vascular Endothelial Growth Factor Receptor (VEGFR) Inhibitors

Since GBM is a highly vascularized tumor, disrupting the tumor vasculature by targeting the VEGF/VEGFR pathway represents an attractive approach to control GBM progression. The VEGF gene family consists of six secreted ligands, including VEGF-A, -B, -C, -D, and placental growth factors (PIGF) 1 and 2 [186]. These ligands bind their corresponding receptors (i.e., VEGFR-1, -2, and -3) and trigger angiogenesis, vasculogenesis, or lymphangiogenesis [186]. VEGF can be blocked using the anti-VEGF mAb bevacizumab or trapped by a soluble decoy receptor such as aflibercept. VEGFR signaling could also be inactivated by an interfering interaction with its ligand by icrucumab [VEGFR-1 inhibitor], ramucirumab [VEGFR-2 inhibitor] or tyrosine kinase activation (e.g., cediranib, sunitinib, pazopanib) [50]. The current clinical trials employing angiogenic VEGFR inhibitors in GBM are listed in Table 2. Although VEGFR inhibitors are being extensively studied, a recent meta-analysis on approximately 2000 GBM patients revealed that an anti-VEGF mAb in combination with standard therapy did not improve the OS of GBM patients compared to standard therapy alone [187]. GBM often develops resistance to vascular inhibitor drugs [188]. Essentially, a better understanding of acquired GBM resistance to VEGF/VEGFR targeted therapies is needed to select an optimal combined therapy to improve therapeutic outcomes. 

##### Platelet-Derived Growth Factor Receptor (PDGFR) Inhibitors

PDGF amplifications are observed in 13% of GBM cases, which makes them the second most frequent somatic alteration in GBM after EGFR [78]. GBM expresses all isoforms of PDGF ligands including PDGF-A, -B, -C, and -D. These growth factors bind their transmembrane receptors PDGFR-α or -β and initiate downstream autophosphorylation events [189] that contribute to various physiological mechanisms of GBM progression including the transformation of glial cells into stem cells, angiogenesis, lymphangiogenesis, and immunosuppression [189,190]. Overexpression of PDGFR in GBM is oftentimes associated with a poor prognosis [191]. Although pre-clinical efficacy of PDGFR inhibitors is compelling, PDGFR inhibitors are facing clinical hurdles in patients since GBM cells often induce a number of signaling pathways such as receptor tyrosine-protein kinase erbB-3 (ERBB3), insulin-like growth factor 1 receptor (IGF1R), and transforming growth factor-beta receptor II (TGFBR2) that provoke resistance to PDGFR abrogation [192,193]. 

#### 4.1.4. Targeting Other Pathways

##### Transforming Growth Factor-Beta (TGF-β) Inhibitors

TGF-β is a pleiotropic cytokine involved in GBM angiogenesis, proliferation, progression, and invasion [194]. High expression of TGF-β is associated with poor prognosis in newly diagnosed GBM patients rather than individuals with recurrent disease [194]. Due to TGF-β′s complex cross-talk with other signaling pathways such as Wnt, Notch, Hippo, MAPK, PI3K-Akt and NF-κB/IKK [195], TGF-β inhibition has not yet gained clinical benefits. In a recent phase Ib/IIa trial, the addition of galunisertib (a small molecule TGF-β inhibitor) with TMZ and radiotherapy in newly diagnosed GBM patients did not improve progression-free survival (PFS) or OS [196]. 

##### Proteasome Inhibitors

Proteasomal inhibition is another approach for GBM treatment intervention [197,198]. The over-expression and enhanced activity of the proteasome are observed in GBM cells following radiotherapy and subsequent proteasome inhibition prevents GBM recurrence [199]. Proteasome inhibitors that have gained access in GBM clinical studies include bortezomib, ixazomib, and marizomib [200]. However, results from bortezomib clinical trial are disappointing as the drug does not cross the BBB [200]. 

##### DNA Damage Response (DDR) Inhibitors

GSCs upregulate various DNA repair proteins such as ataxia telangiectasia mutated (ATM), ataxia telangiectasia and Rad3-related (ATR), or poly (ADP-ribose) polymerase 1 (PARP-1) that facilitate GBM resistance to chemotherapy and/or radiotherapy [201,202]. Therefore, inhibition of these DDR proteins offers an attractive therapeutic approach for GBM patients to overcome resistance to the current standard-of-care. In fact, ATM inhibition defeats DDR-mediated resistance and significantly radio-sensitizes GBM preclinically [203]. The current number of clinical trials with ATM/ATR inhibition in GBM is limited (Table 2). The PARP inhibitors (PARPi) such as olaparib, veliparib, pamiparib are also currently being tested in a number of clinical trials either as single agents or in combination with other therapies (Table 2).

### 4.2. Immunotherapy

#### 4.2.1. Immune Checkpoint Inhibitors (ICIs)

##### Anti-PD-1/PD-L1 Antibodies

A number of newer therapies in development constitute a major area in immunotherapy (Figure 3) [20,21,204,205,206,207]. GBM is an immunosuppressive tumor, with varying degrees of PD-L1 expression in tumor cells in GBM patients ranging from 61% to 88% [208]. Blocking PD-1/PD-L1 interactions unleash anti-tumor immune responses in various cancers such as melanoma. In GBM, although preclinical experience with PD-1/PD-L1 inhibition is quite promising [209,210,211,212], the clinical outcome of PD-1 blockade has been disappointing. For instance, the first large-scale randomized trial in recurrent GBM (CheckMate-143) revealed no significant difference in overall survival between patients receiving bevacizumab or nivolumab (an anti-PD-1 mAb), leading to premature termination of the nivolumab arm [213]. The inability of ICIs to cross the BBB, reduced frequency of immune infiltrates in the GBM TME, and a high level of GBM immunosuppression are considered major contributors of treatment failure for this approach in general [213]. In addition, the TME of PD-1 blockade non-responders is enriched with PTEN mutations regardless of GBM subtype, suggesting that combined targeting of PTEN and PD-1 could provide additive treatment benefits for this lethal disease [214]. Although adjuvant monotherapy of anti-PD-1 mAb failed to generate effective anti-tumor immunity, neoadjuvant PD-1 blockade led to the activation of GBM-specific T cells and downregulation of genes associated with the tumor cell-cycle. Therefore, timing of anti-PD-1/PD-L1 interventions in patients is probably crucial for mediating objective response rates and managing GBM [215].

##### Anti-CTLA-4 Antibody

Cytotoxic T-lymphocyte-associated protein 4 (CTLA-4) is found on antigen-presenting cells (APCs) and Tregs, although the prognostic role of CTLA-4 expression in cancer patients is controversial [216]. A recent study demonstrated that increased expression of CTLA-4 in the GBM TME positively correlated with elevated expression of specific gene signatures in immune cells such as CD8^+^ T cells, Tregs, and macrophages, suggesting a greater immune cell infiltration in tumors with higher CTLA-4 expression [97,217]. Patients with increased levels of CTLA-4 are also likely to beneficially respond to CTLA-4 blockade. The same study concluded that glioma patients with lower CTLA-4 expression have significantly longer OS rates [97]. Relatedly, the safety and efficacy of anti-CTLA4 mAb therapy (e.g., ipilimumab) have been demonstrated in melanoma patients harboring brain metastasis, suggesting promise for individuals with GBM [217,218,219]. Consistent with convincing reports from preclinical studies [217,219], anti-CTLA4 treatment is currently showing encouraging results in clinical trials. In one study in recurrent GBM, ipilimumab in combination with GM-CSF and bevacizumab resulted in partial responses (31%) and stable disease (31%) in select patients [220]. Currently, several clinical trials are running in GBM to determine the safety and efficacy of ipilimumab in combination with drugs such as TMZ, bevacizumab, and other ICIs (Table 3).

##### IDO Inhibition

The less pronounced efficacy of various ICIs (such as anti-PD-1, PD-L1 or CTLA-4 mAbs) in GBM can be attributed, in part, to immunosuppression mediated by alternative pathways such as indoleamine 2,3 dioxygenase (IDO) [221,222]. Indoleamine 2,3 dioxygenase 1 (IDO1) is a rate-limiting enzyme in the kynurenine pathway of tryptophan metabolism [223,224]. Cancer cells activate IDO as a mechanism to limit the bioavailability of tryptophan and, thereby, limit the function of CD8^+^ T and NK cells while also inducing the differentiation of CD4^+^ Tregs [223,225]. The kynurenine pathway is considered a primary resistance mechanism of GBM to chemotherapy and radiation therapy [222], and increased expression of IDO1 in tumor-infiltrating T cells is usually linked to poor overall survival in brain tumor patients [226,227], while IDO loss is associated with reduced recruitment of Tregs in the brain and significantly better patient prognosis [95]. Since the IDO pathway plays a critical role in GBM immunosuppression, interest in IDO pathway inhibition is rapidly increasing. IDO-deficient gliomas recruit antigen-specific CD4^+^ T cells, leading to a significant extension of survival compared to IDO-competent tumors [228]. Anti-tumor efficacy of IDO blockade in vivo is further enhanced by anti-PD-1 and/or radiation therapy [229]. In addition, the combination of IDO inhibition and TMZ/radiation therapy increased tumor cell destruction and survival in mouse GBM models [107,230]. The simultaneous use of IDO inhibition and anti-PD-L1/anti-CTLA4 treatment also led to 100% long-term survivors of mice bearing intracranial gliomas [228]. Currently, several IDO inhibitors are being tested clinically in combination with standard therapies and/or ICIs in GBM (Table 3). 

#### 4.2.2. Oncolytic Viruses (OVs)

Oncolytic viruses (OVs) are a major developmental therapy of interest and have gained success in different cancer types, including GBM [139,205,231,232,233,234,235,236,237,238,239,240,241,242,243]. The recent FDA approval of an oncolytic herpes simplex virus (oHSV) (designated talimogene laherparepvec) has further fueled the field of oncolytic virotherapy. Tumor cells are noticeably distinct from normal cells by adopting behaviors such as increased proliferation and vascularization. OVs selectively infect tumor cells, killing them following replication, while leaving normal cells unscathed. At the same time, OVs trigger a cascade of anti-tumor immune responses [244], including increased tumor infiltration of immune cells [238,245]. OVs including oHSVs have shown promising efficacy in preclinical models of GBM [238,246,247] as well as in GBM patients [248]. In a recently published case study, four previously treated GBM patients received individualized treatment regimens comprised of three OVs (wild-type Newcastle disease virus [NDV], wild-type parvovirus [PV], and wild-type vaccinia virus [VV]). OVs were sequentially administered using the same catheter with a dose of 10^9^ TCID_50_ for each virus in a volume of 10 mL and demonstrated impressive clinical and radiological responses with long-term survival up to 14 years [249]. OV-induced anti-tumor immunity can be further enhanced through OV-mediated expression of various cytokines/chemokines and immunomodulatory molecules [250]. OVs can also be used in combination with standard of care TMZ that produces synergistic anti-tumor effects in various preclinical cancer models including GBM [251,252,253,254,255,256]. However, a recent preclinical combination study (OV+TMZ) in GBM demonstrated conflicting results. Concurrent OV and TMZ therapy antagonized the anti-tumor properties of oncolytic virotherapy [239], indicating that co-applied administration of OV and TMZ represent a failed synergistic strategy as opposed to the pre-clinical benefits observed when TMZ was administered either before or after OV treatment [254,256,257]. Altogether, OVs serve to beneficially alter the TME to increase tumor immunogenicity, and synergize with ICIs [205,258]. Newer clinical studies are aiming to combine OVs and ICIs in order to improve patient outcomes as listed in Table 3 [258].

#### 4.2.3. Therapeutic Vaccines

Vaccines are an active form of immunotherapy that has recently gained interest for GBM treatment [259,260]. The antigens such as EGFR*vIII*, heat shock protein (HSP), and any tumor-derived antigens can be loaded to DCs to incite immune responses against GBM [260]. In a randomized Phase II clinical trial in patients with relapsed EGFR*vIII*^+^ GBM, the EGFR*vIII* vaccine (designated Rindopepimut or CDX-110) delivered intradermally with GM-CSF (NCT01498328) resulted in the induction of EGFR*vIII*-specific immune responses, encouraging PFS and OS, and a significant extension of survival when the vaccine was administered in combination with bevacizumab [261]. The promising results of this trial led to a Phase III trial with Rindopepimut/GM-CSF in patients with newly diagnosed GBM, where all patients received standard-of-care TMZ (NCT01480479). Unfortunately, this Phase III trial was discontinued in early 2016 since Rindopepimut failed to significantly improve survival [262] and emphasizes the importance of identifying alternate and newer vaccine-based strategies to tackle GBM [263].

In contrast to EGFR*vIII* immunization that elicits immune responses to pre-defined tumor target, HSP vaccines offer immunity against a broad range of antigens. Induction of anti-tumor immunity against various antigenic targets is important to help minimize the outgrowth of target null variants, especially for cancer types that have high intra-tumoral heterogeneity like GBM [264]. The most well-known HSP vaccine is heat-shock protein peptide complex-96 (HSPPC-96) [265]. The safety and immunogenicity of HSPPC-96 monotherapy were demonstrated in a Phase I clinical trial in newly diagnosed GBM Patients [265]. HSPPC-96 is currently being tested in two separate Phase II clinical trials; one in combination with TMZ in patients with newly diagnosed GBM (NCT00905060) and the other in combination with bevacizumab in surgically resectable recurrent GBMs (NCT01814813) (Table 3). 

DCs play a central role in linking innate and adaptive anti-tumor immune responses [266]. The principle of dendritic cell vaccines (DCV) is based on the ability of primed DCs to process/present tumor antigens and activate cytotoxic lymphocytes [267]. DCVs are prepared by isolating CD14^+^ monocytes from patient peripheral blood and further culturing cells ex vivo with granulocyte-macrophage colony-stimulating factor (GM-CSF), interleukin 4 (IL-4), and tumor antigens, prior to injecting the cells back into patients [268]. Although interest in DCV is further compelled by an FDA-approved DCV for the treatment of prostate cancer (Sipuleucel-T), most DCV-based clinical trials in GBM are still under phase I and II evaluations. For example, DCVax, an approved DCV for treatment of GBM in Switzerland, is currently being assessed in the US in patients with newly diagnosed GBM (NCT00045968) [269]. In a recent phase III study, DCVax was used alongside standard options and resulted in the extended survival of patients by 8 months compared to the control cohort [269,270]. Personalized neoantigen vaccine has also recently been tested in GBM clinical trials. For instance, in a Phase I/Ib study in newly diagnosed MGMT-unmethylated GBM, patients who did not receive dexamethasone had better neoantigen-specific CD4^+^ and CD8^+^ T cell responses with a higher number of TILs [271].

#### 4.2.4. Adoptive Cell Therapies (ACTs)

In contrast to active immunotherapies such as OVs or anti-cancer vaccines that induce anti-tumor immunity within the host, ACTs are considered a passive form of immunotherapy where T cells are harvested from patients, expanded and activated ex vivo under appropriate conditions, and reinfused back into patients [272]. There are several types of ACT currently under development for GBM treatment, such as lymphokine-activated killer cells, allogenic donor lymphocyte infusion, autologous lymphocytes, tumor-infiltrating lymphocytes (TILs), transgenic T-cell receptor (TCR) T cells, antibody-armed T cells, and chimeric antigen receptor (CAR) T cells [272]. ACTs using autologous TILs and CAR-T cells are the most prevalent strategy being explored in GBM, as evidenced by the number of completed or running clinical trials [272]. ACT by autologous lymphocytes relies on MHC-restricted tumor antigen recognition via T cell receptors (TCRs) [273]. It has also been demonstrated that TILs harvested from GBM patients can recognize and kill autologous tumor cells [274], but recent studies reveal that not all TILs are tumor-specific. Ultimately, TILs that express CD39 (CD39^+^CD8^+^ T cells) are predominantly tumor-reactive [275,276] and these reports suggest that ACT strategies should incorporate CD39^+^CD8^+^ cells over bulk TILs also containing CD39^-^CD8^+^ cells. 

Among ACT-based treatment strategies, CAR-T cell therapy is the furthest along in the clinic [277]. CAR-T cells are engineered to directly recognize tumor-specific antigens (e.g., EGFR*vIII*, HER2, IL-13Rα2, EphA2.) in an MHC-independent manner [273]. As an example, single intravenous infusion of CAR-T cells redirected to EGFR*vIII* antigen in patients with recurrent GBM lead to the loss of EGFR*vIII* in 5/7 patients [278]. Yet, CART-EGFR*vIII* therapy also induced the overexpression of inhibitory molecules such as immune checkpoints in the TME and increased tumor infiltration of immunosuppressive Tregs [278]. Recently, PD1-TILs have been engineered to overcome the inhibitory functions of the PD-1 molecule. PD1-TILs express full-length PD-1 antibody and are currently being tested in GBM patients in an early Phase I clinical trial (NCT03347097). In another study, a recurrent GBM patient received multiple intracranial infusions of IL13Rα2-specific CAR-T cells directly into the resected tumor cavity followed by infusions into the ventricular system over a period of 220 days. This strategy resulted in the regression of all intracranial and spinal tumors [279]. 

#### 4.2.5. Macrophage and NK Cell-Based Immunotherapy

TAMs are another interesting therapeutic target in GBM since these cells are the most abundant infiltrating immune cells in GBM [280] and their immunosuppressive functions promote tumor progression [281]. TAMs can be divided into two major phenotypes: (i) M1-polarized cells that are generally considered anti-tumoral and (ii) M2-like cells that contribute to pro-tumoral activities [282]. The strategies to target TAMs in GBM include: (i) Inhibiting TAM recruitment by preventing interactions between C-C motif chemokine ligand 2 (CCL2) and C-C motif chemokine receptor 2 (CCR2) [283]; (ii) Increasing M1 polarization by disrupting the CD47-SIRPα-SHP-1 signaling pathway [284] or activating CD40 or toll-like receptors (TLRs) [285,286]; and (iii) Depleting TAMs using CSF1R inhibitors [287]. Overall, these aforementioned approaches have demonstrated promise in preclinical GBM studies and are well-tolerated among cancer patients [288]. Nevertheless, the safety and efficacy of macrophage-based immunotherapies need to be confirmed in GBM patients.

In contrast to macrophages, NK cells are the least abundant infiltrating immune cell type in GBM. Interestingly, NK cells contribute to immune surveillance by preventing spontaneous metastasis in GBM [289,290]. Therapeutic approaches that deploy NK cells in GBM are still in early developmental stages, with only one phase I clinical trial exploring the safety and objective response rates following NK cell-based immunotherapy in recurrent GBM patients (NCT04489420). In this study, CYNK-001 cells (i.e., NK cells derived from human placental CD34^+^ cells) will be intravenously infused after a lymphodepleting cyclophosphamide-based chemotherapy or provided intratumorally prior to surgery without lymphodepletion. Although no NK cell-based product has yet received FDA approval, numerous CAR-NK cells have been engineered to target tumor antigens such as EGFR*vIII*, HER2, IL-13Rα2, EphA2, CSPG4, CD133, or CD70 preclinically in GBM [289].

### 4.3. Nanomedicine

The neoplastic vasculature network is typically defective and leaky, which enhances the permeability and retention of nanoparticles in the TME [291,292]. Nanomedicine has been validated to enhance the efficacy of chemotherapies [293] and radiotherapy [294], but the safety and delivery of this approach have always been a major concern since nanoparticles preferentially deposit in the reticuloendothelial tissues of the kidneys, liver, and spleen [295,296]. Despite these issues, various anti-cancer nanoparticle therapies can produce superior efficacy versus non-nanoparticle formulation. For example, paclitaxel (PTX) or doxorubicin when administered as nanoparticles potentiate improved cytotoxicity against GBM compared to their parental compounds [297]. Several clinical trials are evaluating the therapeutic properties of different nanomedicine formulations such as nanoliposome (NCT00734682, NCT00944801, NCT01906385), Spherical Nucleic Acid (SNA) gold nanoparticles (NCT03020017), and nanocells (NCT02766699). To further advance this field in GBM and improve safety, an improved understanding of the long-term stability, biodistribution, and clearance mechanisms of nanoparticles is required.

### 4.4. Photodynamic Therapy (PDT)

Photodynamic therapy (PDT) is a form of light therapy that preferentially damages residual tumor cells following surgical resection [298]. PDT requires molecular oxygen (a photosensitizing [PS] agent) and a light source [299]. Based on their chemical structure, PS agents can be classified as derivatives of chlorin, porphyrin, bacteriochlorin, or phthalocyanine [300]. Ultimately, the PS agent is localized in tumor cells and remains non-toxic until activated by a light source. As the PS agent absorbs photons at a specific wavelength, the transfer of energy converts the PS agent to an excited state and promotes two types of photochemical reactions [301]: (i) Direct interaction between excited PS agents and biochemical molecules within target cells eventually generate cell-destroying free radicles; (ii) The radical anion of an excited PS indirectly interacts with oxygen, leading to the formation of single oxygen molecules that damage mitochondrial DNA and promote mitotic arrest and apoptosis [302,303]. Interestingly, the formation of these oxygen molecules is short-lived (4 micro-seconds) and limited to a maximum 1 μm migration path, which confer minimal harm to surrounding normal cells [304]. A meta-analysis of more than 1,000 GBM patients from several observational studies and three randomized clinical trials (RCTs) [305,306] (NCT01966809) indicate that PDT is safe and significantly improved the quality of life, survival, and delayed tumor relapse in patients (*p* < 0.001) [307]. The combination of PDT with standard therapies (i.e., maximum resection surgery followed by concomitant radio-chemotherapy and adjuvant chemotherapy) is also found to be safe and well-tolerated [308]. A comprehensive review of PDT (including its immunological effects and translational feasibility) has been previously discussed [303]. Although PDT shows favorable outcomes in GBM patients, the field still requires a greater number of pre-clinical studies and RCTs to better confirm safety and efficacy for GBM management. 

### 4.5. Inhibition of Extracellular Vesicles (EVs) and Micro RNA (miRNA)-Based Therapies

Cells in general communicate and sense their environment by employing EVs loaded with nucleic acids, lipids, and proteins [309]. EVs can include exosomes, micro-vesicles, apoptotic bodies, and oncosomes, and are involved in directing cell metabolism and movement [309]. Within the realm of GBM, EVs can be utilized as biomarkers for diagnosis, prognosis and GBM recurrence [310,311]. More specifically, an increase in plasma EV concentration is associated with GBM recurrence [310] while EVs contribute to GBM invasion and recurrence after treatment by transferring therapy-resistant epigenetic materials between GBM cells [312]. Additionally, GBM cells can internalize bevacizumab and release it back into the TME where EVs neutralize the antibody to prevent VEGF-A binding, leading to drug resistance [313]. Therefore, EV inhibition is expected to enhance GBM treatment outcomes. EV production can be prevented by either preventing EV trafficking [312] or lipid metabolism [313]. CCR8 is usually required for EV fusion to cells [313]. As such, the small molecule CCR8 inhibitor R243 (when used in combination with TMZ) demonstrated a significant delay of tumor recurrence [313]. A separate lipid metabolism inhibitor (designated GW4869) also enhanced the cytotoxic effect of bevacizumab in GBM [313]. 

miRNA is the distinct cellular product transferred by EVs between GBM cells [309]. miRNAs are non-coding RNAs that consist of about 22 base pairs and regulate gene expression by binding complementary mRNA sequences to silence translation. miRNA is a major cellular communication scheme in GBM [314] and abnormal miRNA regulation favors tumor growth and invasion [315]. GBM patients have distinct miRNA expression profiles over healthy individuals—with at least 30 miRNAs that can be used as biomarkers [315]. The most frequently up-regulated miRNAs in GBM include miR-21, miR-10b, and miR-25, whereas miR-139, miR-218, and miR-124 are commonly down-regulated in GBM patients [315]. Emerging approaches to target miRNAs include inhibiting their occurrence by administering anti-sense oligonucleotides or therapeutics that silence miRNA expression. For example, inhibition of miR-21 suppresses GBM proliferation by inhibiting EGFR signaling pathway [316], increasing PTEN expression [317], and enhancing chemosensitivity [136,318]. Systemic administration of miR-10b antisense oligonucleotide inhibitors (ASO) also significantly prolonged survival in a xenograft model of mouse GBM [319]. Expression of miRNA-10b and its relation to OS and PFS in glioma patients is likewise currently under investigation (NCT01849952). The exogenous delivery of miR-124 decreases tumor growth and invasion and sensitizes GBM cells to chemotherapy [320]. Lastly, GSCs express different miRNAs over non-stem GBM cells that correlates with patient survival [321]. Some clinical trials consist of miRNA-based therapies and EV inhibitors are currently underway for a variety of cancer types (NCT02580552, NCT03713320, NCT03608631, NCT04167722), but these strategies have not yet been evaluated in GBM patients. 

### 4.6. Targeting Vessel Co-Option and Vascular Mimicry

As discussed above, neovascularization of GBM comprises vessel co-option, angiogenesis, vasculogenesis, endothelial cell trans-differentiation, and vascular mimicry [43]. Among therapies that target GBM tumor vasculature, AAT is the most studied in the clinic. However, GBM treatment with AAT oftentimes increases vascular co-option [322] and results in resistance to AAT [323]. Mechanisms that drive vessel co-option in GBM and non-GBM cancers are poorly understood, although various tumor cell invasion/adhesion pathways are known to be involved as drivers of vessel co-option in glioma [45], such as bradykinin, CXC-chemokine receptor-4 (CXCR4)-binding cytokines, stromal cell-derived factor-1α (SDF1α), interleukin 8 (IL-8), angiopoietin 2 (Ang-2), cell division control protein 42 (CDC42), EGFR*vIII*, mammary-derived growth inhibitor/fatty acid-binding protein 3 (MDGI/FABP3), inositol-requiring enzyme-1α (IRE1α), homologous wingless and Int-1 (Wnt), and oligodendrocyte transcription factor (Olig2) [45,324,325,326]. It has been proposed that sequential treatment of vessel co-option inhibition by LGK974 (a porcupine inhibitor that blocks Wnt secretion) followed by anti-angiogenic therapy could produce synergistic effects that are superior to single treatments [323]. Vascular mimicry is a separate mechanism observed in GBM following AAT-induced resistance [61]. Treatment with AAT enhances a population of CXCR2-positive GBM-stem cells with endothelial-like phenotypes that promote tumor growth. Targeting vascular mimicry in this scenario by blocking the expression of CXCR2 with the compound SB225002 demonstrated significant reduction of tumor burden [61].

## 5. Conclusions

GBM is a devastating disease with an exceedingly poor prognosis and has an expected survival of only 12–15 months [1,2]. Currently approved treatments only manage to increase overall survival by a few months and further research is desperately needed in order to make a significant difference in the progression of the disease. Increasing our understanding of GBM pathogenesis is vital toward developing efficacious and long-lasting therapies. Heterogeneity of GBM is observed at both intra-and inter-tumoral levels, making targeted approaches (including small molecule drugs and immunotherapies) difficult to produce substantial clinical benefits. Therefore, a better understanding of the molecular heterogeneity and immunosuppressive profile of GBM would provide a more comprehensive insight into strategies that can overcome resistance acquired by this lethal disease. While combination therapies in development with the current standard of care options are gaining more attention, it is crucial to assess whether antagonizing properties develop. For example, while the addition of a histone deacetylase inhibitor to TMZ ensures GBM eradication [327], anti-EGFR*vIII* or anti-MAPK strategies potentially abrogate TMZ efficacy by interfering with the regulation of DNA mismatch repair in GBM [328]. Similarly, the combination of OV and TMZ shows a reduced efficacy against GBM compared to either agent alone [239]. It is also needed to investigate sequential administration of different therapies to potentially gain synergistic effects. For example, drugs that target EVs could be administered prior to AATs. Another consideration for improved GBM prognosis and treatment is the timing and method of diagnosis. The BBB is another major obstacle that should be addressed to achieve suitable therapeutic responses in individuals with GBM, particularly for emerging treatment approaches such as OVs, small molecule inhibitors, therapeutic vaccines, miRNA-based treatments, and EVs. It also remains vital to find new therapeutic targets in GBM. Since GSCs play an important role in GBM pathogenesis and therapy resistance, targeting GSCs would offer a unique way to eradicate the disease. Recently, ADAM Like Decysin 1 (ADAMDEC1) was discovered as a novel target in GBM and is overexpressed by GSCs, which regulate stem cell proliferation and sphere formation, and promote tumor growth through an ADAMDEC1-FGFR1-ZEB1 signaling loop [329]. Overall, the rapid growth and aggressive nature of GBM likely demand the necessary focus on developing effective treatments that will work alongside current mainstays of treatment.

## Figures and Tables

**Figure 1 cancers-13-00856-f001:**
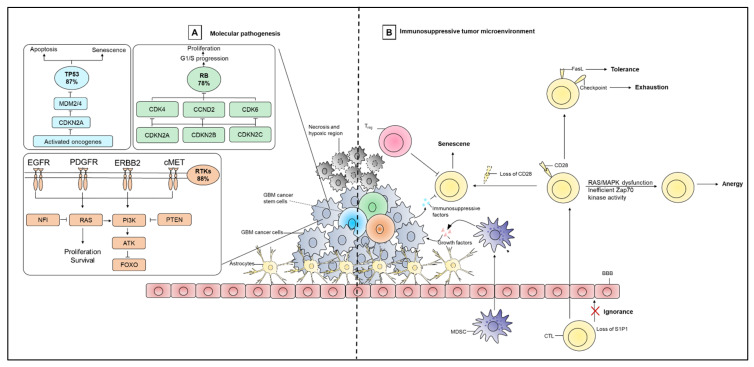
Characteristics of the GBM tumor microenvironment. (**A**) Three dominant molecular alterations in GBM include P53, retinoblastoma (Rb), and receptor tyrosine kinase (RTK) signaling pathways and their corresponding frequencies in GBM. (**B**) The GBM tumor microenvironment (TME) is immunosuppressed mainly due to presence of dysfunctional T cells and myeloid-derived suppressor cells (MDSCs). T cell dysfunction results from different mechanisms that involve senescence, tolerance, anergy, exhaustion, and ignorance. T cells senescence is associated with loss of the co-stimulator CD28, which may occur as a result of telomere damage, presence of Tregs, or TME metabolic stress. Tolerance of T cells is mediated by FasL-induced apoptosis, recruitment of Tregs, and upregulation of factors that limit T cell effector functions. T cell anergy occurs due to RAS/MAPK dysfunction or inefficient Zap70 kinase activity. Co-expression of immune checkpoint molecules such as PD-1, LAG-3 and TIM-3 are also important contributors to T cell exhaustion in GBM. The loss of sphingsosine-1-phosphate receptor 1 (S1P1) is associated with T cell ignorance. MDSCs release immunosuppressive growth factors and other mediators to support growth of cancer cells. Cancer cells can also secret a number of immunosuppressive factors to maintain their overall survival.

**Figure 2 cancers-13-00856-f002:**
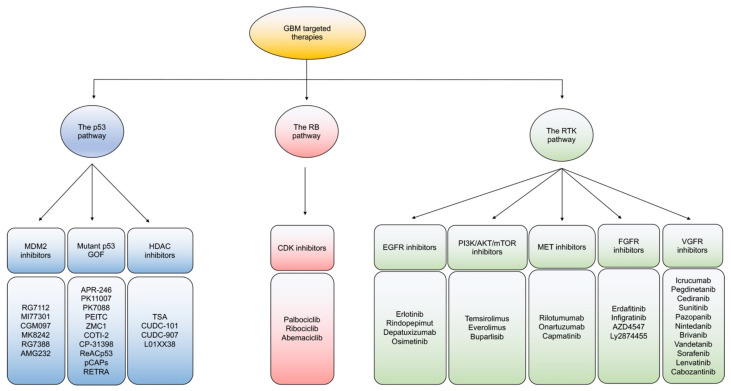
An overview of targeted therapies in GBM. Classification of current targeted therapies in GBM according to the three main signaling pathway alterations of the P53, Rb, and RTK pathways.

**Figure 3 cancers-13-00856-f003:**
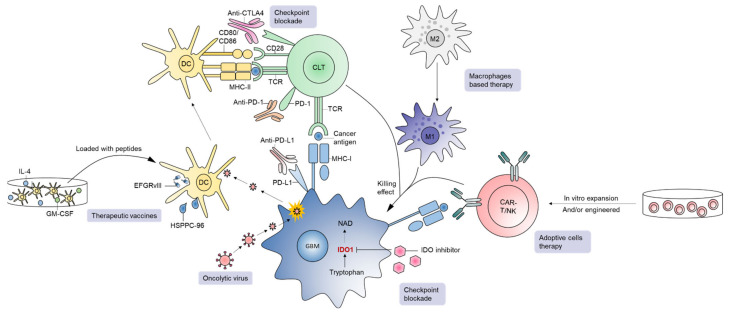
A brief overview of immunotherapies in GBM. Current immunotherapeutic approaches in GBM include checkpoint blockade, oncolytic virus, therapeutic vaccines, adoptive cell therapy, and macrophage-based strategies.

**Table 1 cancers-13-00856-t001:** Glioblastoma (GBM) molecular subtypes.

GBM Subtype	Expression of Signature Genes	Predominant Marker	Distinct Neural Cell Types
Proneural	PDGFRA alternation IDH1 point mutation	Proneural markers: SOX, DCX, DLL3, ASCL1, TCF4 and Oligodendrocytic development markers: *PDGFRA*, *NKX2-2* and *OLIG2*	Oligodendrocyte
Mesenchymal	Lower NF1 expression · NF1 and PTEN co-mutation High expression of TRADD, RELB, TNFRSF1A	Mesenchymal and astrocytic markers CD44, MERTK	Astroglial
Classical	Chromosome 7 amplification paired with chromosome 10 loss High level of EGFR amplification Point or vIII EGFR mutation Lack of TP53 mutation	Neural precursor and stem cell markers NES, NOTCH3, JAG1, LFNG, SMO, GAS1, GLI2	Murine astrocytes
Neural	Neuron projection, and axon and synaptic transmission	Neuron markers such as NEFL, GABRA1, SYT1 and SLC12A5.	Neuron, oligodendrocytes and astrocytes

**Table 2 cancers-13-00856-t002:** Current on-going clinical trials of targeted therapies in GBM within the last 5 years.

Targeted Pathway	Targeted Therapy	Drug Name	In Combination	Condition	Phase	N	NCT (Accessed On)
The p53 pathway	MDM2 inhibitors	AMG232	Radiation	Newly Diagnosed and recurrent GBM	I	86	NCT03107780 (04/09/2020)
		RG7388 (Idasanutlin)	Radiation	Newly Diagnosed GBM Without MGMT Promoter Methylation	I/IIa	350	NCT03158389 (04/09/2020)
	HDAC inhibitors	SAHA (Vorinostat)	Radiation Pembrolizumab TMZ	Newly Diagnosed GBM	I	32	NCT03426891 (04/09/2020)
			Isotretinoin TMZ	Recurrent GBM	I/II	135	NCT00555399 (04/09/2020)
			Radiation TMZ	Newly Diagnosed GBM	I/II	125	NCT00731731 (04/09/2020)
			Bevacizumab	Recurrent GBM	II	48	NCT01738646 (04/09/2020)
		CUDC-907 (Fimepinostat)	Surgery	Recurrent GBM	Early I	30	NCT03893487 (04/09/2020)
The Rb pathway	CDK inhibitor	PD-332991 (Palbociclib)	Radiation	Newly Diagnosed GBM Without MGMT Promoter Methylation	I/IIa	350	NCT03158389 (04/09/2020)
		LEE011 (Ribociclib)	Everolimus	Recurrent GBM	Early I	24	NCT03834740 (04/09/2020)
				Preoperative GBM	Early I	48	NCT02933736 (04/09/2020)
		LY2835219 (Abemaciclib)	Bevacizumab	Recurrent GBM	Early I	10	NCT04074785 (04/09/2020)
				Recurrent GBM	II	42	NCT02981940 (04/09/2020)
			LY3214996	Recurrent GBM	Early I	50	NCT04391595 (04/09/2020)
			TMZ	GBM	II	280	NCT02977780 (04/09/2020)
The RTK pathway	EGFR inhibitors	OSI-774 (Erlotinib)		Relapsed/Refractory GBM	I/II	11	NCT00301418 (04/09/2020)
			Bevacizumab TMZ	Newly Diagnosed GBM	II	115	NCT00720356 (04/09/2020)
			Sorafenib	Progressive or recurrent GBM	II	56	NCT00445588 (04/09/2020)
		Cetuximab	Mannitol Radiation	Relapsed/Refractory GBM	II	37	NCT02800486 (04/09/2020)
			Mannitol	Newly Diagnosed GBM	I/II	33	NCT02861898 (04/09/2020)
		AZD9291 (Osimertinib)	Fludeoxyglucose F-18 PET	Recurrent GBM	II	12	NCT03732352 (04/09/2020)
		Nimotuzumab	Radiation TMZ	Newly Diagnosed GBM	II	39	NCT03388372 (04/09/2020)
				Newly Diagnosed GBM	III	150	NCT00753246 (04/09/2020)
		CDX-110 (Rindopepimut)	TMZ	Newly Diagnosed, Surgically Resected, EGFRvIII-positive GBM	III	745	NCT01480479 (04/09/2020)
			Radiation TMZ	Newly Diagnosed GBM	II	82	NCT00458601 (04/09/2020)
		ABT-414 (Depatuxizumab)	TMZ	Recurrent GBM	II	266	NCT02343406 (04/09/2020)
			Radiation TMZ	GBM	I	202	NCT01800695 (04/09/2020)
			Radiation TMZ	Newly Diagnosed GBM With EGFR Amplification	III	691	NCT02573324 (04/09/2020)
			Radiation TMZ	Newly diagnosed or recurrent GBM	I/II	53	NCT02590263 (04/09/2020)
	PI3K/AKT/mTOR inhibitors	CCI-779 (Temsirolimus)	Sorafenib Tosylate	Recurrent GBM	I/II	115	NCT00329719 (04/09/2020)
			Sorafenib Tosylate	Recurrent GBM	I/II	92	NCT00335764 (04/09/2020)
			Perifosine	Recurrent GBM	I	10	NCT02238496 (04/09/2020)
			TMZ	Recurrent GBM	II	266	NCT02343406 (04/09/2020)
		RAD001 (Everolimus)	Ribociclib	Recurrent GBM	Early I	24	NCT03834740 (04/09/2020)
			Radiation TMZ	Newly diagnosed GBM	I/II	122	NCT00553150 (04/09/2020)
			Radiation TMZ	Newly diagnosed GBM	I/II	279	NCT01062399 (04/09/2020)
			Sorafenib	Recurrent GBM	I/II	118	NCT01434602 (04/09/2020)
		BKM120 (Buparlisib)	Bevacizumab	Relapsed/Refractory GBM	I/II	88	NCT01349660 (04/09/2020)
			Radiation TMZ	Newly diagnosed GBM	I	38	NCT01473901 (04/09/2020)
			Lomustine or Carboplatin	Recurrent GBM	Ib/II	35	NCT01934361 (04/09/2020)
				Recurrent GBM	II	65	NCT01339052 (04/09/2020)
	MET inhibitors	AMG 102 (Rilotumumab)	Bevacizumab	Recurrent GBM	II	36	NCT01113398 (04/09/2020)
		(RO5490258) Onartuzumab	Bevacizumab	Recurrent GBM	II	135	NCT01632228 (04/09/2020)
		INC280 (Capmatinib)	Bevacizumab	GBM	I	65	NCT02386826 (04/09/2020)
	FGFR inhibitors	Infigratinib		Recurrent GBM	Early I	20	NCT04424966 (04/09/2020)
		AZD4547		GBM with FGFR-TACC gene fusion	I/II	14	NCT02824133 (01/02/2021)
	VEFGR inhibitors	Cediranib	Bevacizumab Olaparib	Recurrent GBM	II	70	NCT02974621 (04/09/2020)
		Sunitinib	Lomustine	Recurrent GBM	II/III	100	NCT03025893 (04/09/2020)
		Pazopanib	Topotecan	Recurrent GBM	II	35	NCT01931098 (04/09/2020)
			N/A	Newly diagnosed GBM	I/II	51	NCT02331498 (04/09/2020)
		Vandetanib	Sirolimus	Recurrent GBM	I	33	NCT00821080 (04/09/2020)
		Sorafenib	Everolimus	Recurrent GBM	I/II	118	NCT01434602 (04/09/2020)
		Lenvatinib	Pembrolizumab	GBM	II	600	NCT03797326 (04/09/2020)
		Cabozantinib	N/A	Recurrent GBM	II	10	NCT02885324 (04/09/2020)
Other pathways	TGF-β inhibitors	BCA101	Pembrolizumab	GBM	I	292	NCT04429542 (04/09/2020)
		LY2157299 (Galunisertib)	Lomustine	Recurrent GBM	II	180	NCT01582269 (04/09/2020)
	Proteasome inhibitors	Bortezomib	TMZ	Recurrent GBM with Unmethylated MGMT Promoter to TMZ	1B/II	63	NCT03643549 (04/09/2020)
			Bevacizumab TMZ	Recurrent GBM	I	12	NCT01435395 (04/09/2020)
		Ixazomib	N/A	GBM	Early phase I	3	NCT02630030 (04/09/2020)
		Marizomib	Radiation TMZ	Newly diagnosed GBM	III	750	NCT03345095 (04/09/2020)
			Bevacizumab	GBM	I/II	121	NCT02330562 (04/09/2020)
			Radiation TMZ Optune	Newly diagnosed GBM	IB	66	NCT02903069 (04/09/2020)
			ABI-009 (Nab-Rapamycin)	Newly diagnosed GBM	II	56	NCT03463265 (04/09/2020)
	ATM inhibitor	AZD1390	Radiation	Newly diagnosed and recurrent GBM	I	132	NCT03423628 (01/02/2021)
	PARP inhibitor	Veliparib	Radiation TMZ	GBM	I/II	66	NCT01514201 (01/02/2021)
			Radiation TMZ	GBM	II	115	NCT03581292 (01/02/2021)
			TMZ	Newly Diagnosed GBM With MGMT Promoter Hypermethylation	II/III	440	NCT02152982 (01/02/2021)
		Olaparib	Bevacizumab Cediranib	Recurrent GBM	II	70	NCT02974621 (01/02/2021)
			Radiation Pamiparib TMZ	Newly diagnosed and recurrent GBM	I	30	NCT04614909 (01/02/2021)
			N/A	GBM	II	145	NCT03212274 (01/02/2021)
		Pamiparib	Radiation TMZ	Newly diagnosed and recurrent GBM	I/II	116	NCT03150862 (01/02/2021)
			TMZ	Recurrent GBM	I/II	100	NCT03914742 (01/02/2021)
			TMZ	Recurrent GBM	I	78	NCT03749187 (01/02/2021)

N—Number of participants, NCT—National Clinical Trial, MDM2—Mouse double minute 2 homolog, HDAC—Histone deacetylase, CDK—Cyclin-dependent kinase, RB—Retinoblastoma, RTK—Receptor Tyrosine kinase, PI3K/AKT/mTOR—Phosphatidylinositol-3-kinase/AKT/mammalian target of rapamycin, MET—Mesenchymal Epithelial Transition, FGFR—Fibroblast growth factor receptor, VEFGR—Vascular Endothelial Growth Factor receptor, TGF-β—Transforming growth factor beta, GBM—Glioblastoma Multiforme,. MGMT- O^6^-Methylguanine-DNA Methyltransferase, TMZ—Temozolomide, ATM—ataxia telangiectasia mutated, PARP—poly (ADP-ribose) polymerase 1.

**Table 3 cancers-13-00856-t003:** Clinical status of immunotherapies in GBM within the last 5 years (since 2015).

Immunotherapies	Drug Name	In Combination	Condition	Phase	N	NCT (Accessed on)
Anti-PD-1	Spartalizumab	MBG453	Recurrent GBM	I	15	NCT03961971 (11/09/2020)
		BLZ945	Advanced/metastatic/recurrent IDH wild-type GBM	I/II	200	NCT02829723 (11/09/2020)
	INCMGA00012	INCAGN01876, stereotactic radiosurgery	Recurrent GBM	II	32	NCT04225039 (11/09/2020)
		Bevacizumab, radiation	Recurrent GBM	II	55	NCT03532295 (11/09/2020)
	Nivolumab	BMS-986016	Recurrent GBM	I	63	NCT02658981 (11/09/2020)
		Dendritic vaccine	GBM	I	6	NCT02529072 (11/09/2020)
		Bevacizumab	Recurrent GBM	II	90	NCT03452579 (11/09/2020)
		Ipilimumab, Radiation	Newly Diagnosed, MGMT Unmethylated Glioblastoma	II	24	NCT03367715 (11/09/2020)
		TMZ	Newly Diagnosed Elderly Patients With GBM	II	102	NCT04195139 (11/09/2020)
		Ipilimumab, NovoTTF200A (Optune)	Recurrent GBM	II	60	NCT03430791 (11/09/2020)
		Ad-RTS-hIL-12, Veledimex	Recurrent or Progressive GBM	I	21	NCT03636477 (11/09/2020)
		Ipilimumab	Recurrent GBM	I	6	NCT03233152 (11/09/2020)
		Ipilimumab, TMZ	Newly Diagnosed GBM	I	32	NCT02311920 (11/09/2020)
		Bevacizumab, Re-irradiation	Recurrent MGMT Methylated GBM	II	94	NCT03743662 (11/09/2020)
		Ipilimumab	Recurrent GBM with elevated mutational	II	37	NCT04145115 (11/09/2020)
		BMS-986205, TMZ, Radiation	Newly diagnosed GBM	I	30	NCT04047706 (11/09/2020)
		IL13Ralpha2-CRT T cells, Ipilimumab	Recurrent GBM	I	60	NCT04003649 (11/09/2020)
		Ipilimumab	Newly Diagnosed, MGMT Unmethylated GBM	II/III	485	NCT04396860 (11/09/2020)
		TMZ, radiation	Newly Diagnosed, MGMT Methylated GBM	III	693	NCT02667587 (11/09/2020)
		TMZ	Newly Diagnosed, MGMT Unmethylated GBM	III	550	NCT02617589 (11/09/2020)
		Bevacizumab	Recurrent GBM	II	40	NCT03890952 (11/09/2020)
		NeoVax, Ipilimumab	Newly Diagnosed, MGMT Unmethylated GBM	I	3	NCT03422094 (11/09/2020)
		Therapeutic vaccine EO2401	Progressive Glioblastoma	Ib/IIa	32	NCT04116658 (11/09/2020)
		N/A	IDH-Mutant GBM with and without hypermutator phenotype	II	95	NCT03718767 (11/09/2020)
		Ipilimumab	Recurrent GBM	I	45	NCT04323046 (11/09/2020)
	Pembrolizumab	SurVaxM, Sargramostim, Montanide ISA51	Recurrent GBM	II	51	NCT04013672 (11/09/2020)
		Radiation, TMZ, NeoAntigen vaccine	MGMT Unmethylated, Newly Diagnosed GBM	I	56	NCT02287428 (11/09/2020)
		Bevacizumab, radiation	Recurrent GBM	II	60	NCT03661723 (11/09/2020)
		HSPPC-96, TMZ	Newly Diagnosed GBM	II	108	NCT03018288 (11/09/2020)
		N/A	Newly Diagnosed GBM	II	56	NCT03899857 (11/09/2020)
		Laser Interstitial Thermotherapy	Recurrent GBM	I/II	34	NCT03277638 (11/09/2020)
		TTF	Newly Diagnosed GBM	II	29	NCT03405792 (11/09/2020)
		TTAC-0001	Recurrent GBM	I	9	NCT03722342 (11/09/2020)
		EGFRvIII-CAR T Cells	Newly diagnosed, MGMT unmethylated GBM	I	7	NCT03726515 (11/09/2020)
		Vorinostat, TMZ, radiation	Newly Diagnosed GBM	I	32	NCT03426891 (11/09/2020)
		Ferumoxytol	Newly Diagnosed GBM	II	45	NCT03347617 (11/09/2020)
		IMA950/Poly-ICLC	Recurrent GBM	I/II	24	NCT03665545 (11/09/2020)
		Dendritic Cell Tumor Cell Lysate Vaccine	Recurrent or progressive GBM	I	40	NCT04201873 (11/09/2020)
		N/A	Recurrent GBM	II	20	NCT02337686 (11/09/2020)
		Oncolytic Polio/Rhinovirus Recombinant (PVSRIPO)	Recurrent GBM	I	10	NCT04479241 (11/09/2020)
		Radiation, TMZ	Newly Diagnosed GBM	II	90	NCT03197506 (11/09/2020)
		N/A	Recurrent or progressive GBM	I	35	NCT02852655 (11/09/2020)
		Adenovirus (DNX-2401)	Recurrent GBM	II	49	NCT02798406 (11/09/2020)
		Lenvatinib	GBM	II	600	NCT03797326 (11/09/2020)
		Cyclophosphamide, fludarabine, aldesleukin, TIL	Progressive GBM	II	332	NCT01174121 (11/09/2020)
		Cyclophosphamide, fludarabine, aldesleukin, TCR	GBM	II	270	NCT03412877 (11/09/2020)
		MRI-guided Laser Ablation	Recurrent GBM	I/II	58	NCT02311582 (11/09/2020)
		N/A	Recurrent GBM With a Hypermutator Phenotype	Pilot	44	NCT02658279 (11/09/2020)
		Radiation, Bevacizumab	Recurrent GBM	I	32	NCT02313272 (11/09/2020)
		Radiation, TMZ	GBM	I/II	50	NCT02530502 (11/09/2020)
	Cemiplimab	Ad-RTS-hIL-12, Veledimex	Recurrent or progressive GBM	II	36	NCT04006119 (11/09/2020)
		INO-5401 and INO-9012	Newly Diagnosed GBM	I/II	52	NCT03491683 (11/09/2020)
Anti-PD-L1	Atezolizumab	Radiation, TMZ	Newly Diagnosed GBM	I/II	80	NCT03174197 (11/09/2020)
		Ipatasertib	GBM	I/II	51	NCT03673787 (11/09/2020)
		Radiation	MGMT unmethylated GBM	I/II	350	NCT03158389 (11/09/2020)
		D2C7-IT	Recurrent GBM	I	18	NCT04160494 (11/09/2020)
	Avelumab	MRI-guided LITT therapy	Recurrent GBM	I	30	NCT03341806 (11/09/2020)
		Hypofractionated radiation therapy	IDH mutant GBM	II	43	NCT02968940 (11/09/2020)
		VXM01	Progressive GBM	I/II	30	NCT03750071 (11/09/2020)
		TMZ	Newly Diagnosed GBM	II	30	NCT03047473 (11/09/2020)
		Axitinib	Recurrent GBM	II	52	NCT03291314 (11/09/2020)
	Durvalumab	Bevacizumab, radiation	GBM	II	159	NCT02336165 (11/09/2020)
		Hypofractionated stereotactic radiation therapy	Recurrent GBM	I/II	112	NCT02866747 (11/09/2020)
		Tremelimumab	Recurrent GBM	II	36	NCT02794883 (11/09/2020)
Anti-CLTA4	Tremelimumab	Durvalumab	Recurrent GBM	II	36	NCT02794883 (11/09/2020)
	Ipilimumab	Nivolumab, Radiation	Newly Diagnosed, MGMT Unmethylated Glioblastoma	II	24	NCT03367715 (11/09/2020)
		Nivolumab, NovoTTF200A (Optune)	Recurrent GBM	II	60	NCT03430791 (11/09/2020)
		Nivolumab	Recurrent GBM	I	6	NCT03233152 (11/09/2020)
		Nivolumab	Recurrent GBM with elevated mutational	II	37	NCT04145115 (11/09/2020)
		Nivolumab, TMZ	Newly Diagnosed GBM	I	32	NCT02311920 (11/09/2020)
		IL13Ralpha2-CRT T cells, Nivolumab	Recurrent GBM	I	60	NCT04003649 (11/09/2020)
		Nivolumab	Newly Diagnosed, MGMT Unmethylated GBM	II/III	485	NCT04396860 (11/09/2020)
		NeoVax, Nivolumab	Newly Diagnosed, MGMT Unmethylated GBM	I	3	NCT03422094 (11/09/2020)
		Nivolumab	Recurrent GBM	I	45	NCT04323046 (11/09/2020)
Anti-IDO1	BMS-986205 (Linrodostat)	Nivolumab, TMZ, Radiation	Newly diagnosed GBM	I	30	NCT04047706 (11/09/2020)
	Indoximod	TMZ, radiation, cyclophosphamide, etoposide	Newly diagnosed GBM	I	81	NCT02502708 (11/09/2020)
		TMZ, bevacizumab, stereotactic radiation	GBM	I/II	160	NCT02052648 (11/09/2020)
		TMZ, radiation, cyclophosphamide, etoposide, lomustine	Progressive GBM	II	140	NCT04049669 (11/09/2020)
Oncolytic viruses	TG6002	5-flucytosine	Recurrent GBM	I/II	78	NCT03294486 (11/09/2020)
	DNX-2440	N/A	Recurrent GBM	I	24	NCT03714334 (11/09/2020)
	DNX-2401	IFNg	Recurrent GBM	I	37	NCT02197169 (11/09/2020)
		Pembrolizumab	Recurrent GBM	II	49	NCT02798406 (11/09/2020)
		TMZ	Recurrent GBM	I	31	NCT01956734 (11/09/2020)
		N/A	Recurrent GBM	I	36	NCT03896568 (11/09/2020)
	C134	N/A	Recurrent GBM	I	24	NCT03657576 (11/09/2020)
	M032	N/A	Recurrent GBM	I	36	NCT02062827 (11/09/2020)
	G207	Radiation	Recurrent GBM	II	30	NCT04482933 (11/09/2020)
		N/A	Recurrent GBM	I	15	NCT03911388 (11/09/2020)
		N/A	Recurrent or progressive GBM	I	12	NCT02457845 (11/09/2020)
Therapeutic vaccines	Percellvac	N/A	Newly diagnosed GBM	I	10	NCT02709616 (11/09/2020)
	Percellvac2	N/A	Newly diagnosed GBM	I	10	NCT02808364 (11/09/2020)
	GNOS-PV01	Plasmid encoded IL-12	Newly diagnosed, unmethylated GBM	I	6	NCT04015700 (11/09/2020)
	VBI-1901	N/A	Recurrent GBM	I/II	38	NCT03382977 (11/09/2020)
	MTA-based Personalized Vaccine	TTF, Poly-ICLC	Newly diagnosed GBM	I	20	NCT03223103 (11/09/2020)
	Autologous DCV	TMZ	GBM	II	28	NCT04523688 (11/09/2020)
	UCPVax	N/A	GBM	I/II	28	NCT04280848 (11/09/2020)
	NeoAntigen vaccine	Radiation, TMZ, Pembrolizumab	MGMT Unmethylated, Newly Diagnosed GBM	I	56	NCT02287428 (11/09/2020)
	Dendritic Cell Tumor Cell Lysate Vaccine	Pembrolizumab, Poly ICLC	Recurrent or progressive GBM	I	40	NCT04201873 (11/09/2020)
		TMZ, radiation, bevacizumab	Newly diagnosed or recurrent GBM	I	39	NCT02010606 (11/09/2020)
	NeoVax	Nivolumab, Ipilimumab	Newly Diagnosed, MGMT Unmethylated GBM	I	3	NCT03422094 (11/09/2020)
	DC/tumor cell fusion vaccine	IL-12, TMZ	Newly diagnosed GBM	I/II	10	NCT04388033 (11/09/2020)
	Malignant Glioma Tumor Lysate-Pulsed Autologous DCV	N/A	Recurrent GBM	I	20	NCT03360708 (11/09/2020)
		TMZ	Newly diagnosed GBM	I	21	NCT01957956 (11/09/2020)
		0.2% resiquimod, polyICLC	GBM	2	60	NCT01204684 (11/09/2020)
	Autologous, tumor lysate-loaded, mature DCs	TMZ, radiation	Newly diagnosed GBM	II	136	NCT03395587 (11/09/2020)
	Autologous DCV	TMZ, radiation	Newly diagnosed GBM	I/II	20	NCT02649582 (11/09/2020)
	Autologous DCs loaded with autogeneic glioma stem-like cells (A2B5+)	Surgery, chemotherapy, and radiotherapy.	GBM	II	100	NCT01567202 (11/09/2020)
	EO2401	N/A	Progressive or first recurrent GBM	I/II	32	NCT04116658 (11/09/2020)
	SurVaxM	TMZ, Montanide ISA 51 VG, Sargramostim	Newly diagnosed GBM	II	64	NCT02455557 (11/09/2020)
	CMV RNA-Pulsed Dendritic Cells	Tetanus-Diphtheria Toxoid Vaccine	Newly diagnosed GBM	II	120	NCT02465268 (11/09/2020)
		Tetanus-Diphtheria Toxoid Vaccine	Recurrent GBM	I	11	NCT03615404 (11/09/2020)
	HSPPC-96	Pembrolizumab, TMZ	Newly diagnosed GBM	II	108	NCT03018288 (11/09/2020)
	Autologous DCV	metronomic cyclophosphamide	Recurrent GBM	I/II	25	NCT03879512 (11/09/2020)
	CMV-specific dendritic cell vaccine	TMZ, Tetanus-Diphtheria Toxoid, GM-CSF	Newly diagnosed Unmethylated GBM	II	48	NCT03927222 (11/09/2020)
	DEN-STEM	TMZ	IDH wild-type, MGMT methylated GBM	II/III	60	NCT03548571 (11/09/2020)
	ADCV01	N/A	GBM	II	24	NCT04115761 (11/09/2020)
	VXM01	Avelumab	Progressive GBM	I/II	30	NCT03750071 (11/09/2020)
	IMA950	Pembrolizumab/Poly-ICLC	Recurrent GBM	I/II	24	NCT03665545 (11/09/2020)
	ADCTA-SSI-G1	Bevacizumab	Recurrent GBM	III	118	NCT04277221 (11/09/2020)
	AV-GBM-1	N/A	Newly diagnosed GBM	II	55	NCT03400917 (11/09/2020)
	Human CMV pp65-LAMP mRNA-pulsed autologous DCs	Variliumab	Newly diagnosed GBM	II	112	NCT03688178 (11/09/2020)
Adoptive cell therapy	PD1-TIL	N/A	GBM	I	40	NCT03347097 (11/09/2020)
	TIL	Cyclophosphamide, fludarabine, aldesleukin, pembrolizumab	Progressive GBM	II	332	NCT01174121 (11/09/2020)
	Autologous T-Cells Express TCRs Reactive Against Mutated Neoantigens	Cyclophosphamide, fludarabine, aldesleukin, pembrolizumab	GBM	II	270	NCT03412877 (11/09/2020)
	EGFRvIII CAR-T	Pembrolizumab	Newly diagnosed, MGMT unmethylated GBM	I	7	NCT03726515 (11/09/2020)
	B7-H3 CAR-T	TMZ	Recurrent GBM	I	12	NCT04385173 (11/09/2020)
	CD147 CAR-T	N/A	Recurrent GBM	I	31	NCT04045847 (11/09/2020)
	Chlorotoxin (EQ)-CD28-CD3zeta-CD19t-expressing CAR-T	N/A	Recurrent GBM	I	36	NCT04214392 (11/09/2020)
	IL13Ralpha2 CAR-T	Nivolumab, Ipilimumab	Recurrent GBM	I	60	NCT04003649 (11/09/2020)
	NKG2D-based CAR-T	N/A	Recurrent GBM	I	10	NCT04270461 (11/09/2020)
	IL13Ralpha2-specific hinge-optimized 41BB-co-stimulatory CAR Truncated CD19-expressing Autologous T-Lymphocytes	N/A	Recurrent GBM	I	92	NCT02208362 (11/09/2020)
Macrophage based therapy	BLZ945	Spartalizumab	Advanced/metastatic/recurrent IDH wild-type GBM	I/II	200	NCT02829723 (11/09/2020)
	APX005M	N/A	GBM	I	45	NCT03389802 (11/09/2020)
NK cell therapy	NK-92/5.28.z Cells	N/A	Recurrent HER2-positive GBM	I	30	NCT03383978 (11/09/2020)

N—Number of participants, NCT—National Clinical Trial, MRI—Magnetic resonance imaging, IDH—Isocitrate dehydrogenase, N/A—not applicant, PD-1—program death 1, PD-L1—program death ligand 1, IDO-1—Indoleamine 2,3-dioxygenase, TTF—NovoTTF-100A System or Optune, EGFR—epidermal growth factor receptor, TIL—tumor infiltrating lymphocytes, CAR-T- Chimeric antigen receptor T cells, NK- Natural killer cells.

## Data Availability

Not applicable.

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
