# Peer review of "Pathogenetic Features and Current Management of Glioblastoma"

_cancers, 2021, doi:10.3390/cancers13040856_

Round 1

Reviewer 1 Report

Authors have arranged an extensive review briefly depicting the current understanding of GBM’s pathogenetic highlights that advance treatment resistance. Moreover authors also outline a few novel and promising focused on targeted agents currently under development for GBM patients and their current clinical status.

The article may have the high scientific impact as it rises the important issue of changes the immunotherapeutic approaches and tumor microenvironment, which is believed to promote tumor growth and metastasis and a better understanding of the molecular heterogeneity and immunosuppressive profile of GBM would provide a more comprehensive insight on into strategies that can overcome resistance acquired by GBM. The topic is important as it rises the current targeting agent for GBM treatment. The manuscript is well written, clear and easy to read. It is supported by well-prepared figures. Title and abstract are accurate. The topics covered in this review are well and precisely discussed, and references are well cited. I suggest accepting manuscript at present form.

Author Response

We sincerely thank the reviewers for evaluating this manuscript. Please find below point-by-point responses to each comment. Editorial formatting may change the line numbers (from what we indicated below) in the word version of the revised manuscript, thus, we request the reviewers to check the PDF version of the revised manuscript.

REVIEWER 1 EVALUATION:

Authors have arranged an extensive review briefly depicting the current understanding of GBM’s pathogenetic highlights that advance treatment resistance. Moreover, authors also outline a few novel and promising focused on targeted agents currently under development for GBM patients and their current clinical status.

Response: We thank the reviewer for positively summarizing this article.

The article may have the high scientific impact as it rises the important issue of changes the immunotherapeutic approaches and tumor microenvironment, which is believed to promote tumor growth and metastasis and a better understanding of the molecular heterogeneity and immunosuppressive profile of GBM would provide a more comprehensive insight on into strategies that can overcome resistance acquired by GBM. The topic is important as it rises the current targeting agent for GBM treatment. The manuscript is well written, clear and easy to read. It is supported by well-prepared figures. Title and abstract are accurate. The topics covered in this review are well and precisely discussed, and references are well cited. I suggest accepting manuscript at present form.

Response: We appreciate the reviewer for providing positive feedback.

Reviewer 2 Report

The review entitled “Pathogenetic Features and Current Management of Glioblastoma”, by the authors H. M Nguyen et al. represents a very good and interesting summary in the field of glioblastoma.

Please consider the following comments: 

Lines 120-121 There are also other factors involved in brain tumor angiogenesis which were not mentioned here like HGF, bFGF, PDGF, TGF-β, MMPs, and angiopoietins. Please add it and explain their role in angiogenesis.  

Lines 311-318 Please include and add few/sentence about new paper concerning clinical trials where patients were treated with TTFields/RT/TMZ followed by adjuvant TMZ/TTFields

  1. Boxtein et al. Concurrent Tumor Treating Fields (TTFields) and Radiation Therapy for Newly Diagnosed Glioblastoma: A Prospective Safety and Feasibility Study, Front Oncol. 2020

Lines 761-782

Please comment this study in the text:

  1. Dupont et al. INtraoperative photoDYnamic Therapy for GliOblastomas (INDYGO): Study Protocol for a Phase I Clinical Trial, Neurosurgery 2019.

Line 776 three randomized clinical trials………………Which one? Add the numer of clinical trial or appropriate literature.

Author Response

We sincerely thank the reviewers for evaluating this manuscript. Please find below point-by-point responses to each comment. Editorial formatting may change the line numbers (from what we indicated below) in the word version of the revised manuscript, thus, we request the reviewers to check the PDF version of the revised manuscript.

REVIEWER 2 EVALUATION:

The review entitled “Pathogenetic Features and Current Management of Glioblastoma”, by the authors H. M Nguyen et al. represents a very good and interesting summary in the field of glioblastoma.

Response: We thank the reviewer for finding this review “very good and interesting”.

Please consider the following comments: 

Lines 120-121 - There are also other factors involved in brain tumor angiogenesis which were not mentioned here like HGF, bFGF, PDGF, TGF-β, MMPs, and angiopoietins. Please add it and explain their role in angiogenesis.  

Response: Thank you for this comment. We have briefly (due to space constraint) listed several angiogenic factors such as bFGF, PDGF, TGF-b, angiopoietins and their role in tumor angiogenesis to lines 121-124. In addition, we have included HGF and MMPs in the pseudopalisade section (line 130).

Lines 311-318 - Please include and add few/sentence about new paper concerning clinical trials where patients were treated with TTFields/RT/TMZ followed by adjuvant TMZ/TTFields

  1. Boxtein et al. Concurrent Tumor Treating Fields (TTFields) and Radiation Therapy for Newly Diagnosed Glioblastoma: A Prospective Safety and Feasibility Study, Front Oncol. 2020

Response: We appreciate the reviewer for this suggestion. This study has been added to lines 363-368.

Lines 761-782 - Please comment this study in the text:

  1. Dupont et al. INtraoperative photoDYnamic Therapy for GliOblastomas (INDYGO): Study Protocol for a Phase I Clinical Trial, Neurosurgery 2019.

Response: The INDYGO is a very interesting clinical trial assessing the feasibility of intraoperative PDT early after surgical resection of GBM. We have added this study to lines 928-931.

Line 776 - three randomized clinical trials………………Which one? Add the number of clinical trial or appropriate literature.

Response: Three randomized clinical trials actually represent a meta-analytical study of more than 1,000 GBM patients (along with several observational studies). We added NCT number and references of these trials to line 927.

Reviewer 3 Report

The manuscript by Nguyen et al is a review article that aims to overview pathological and genetic features of glioblastoma (GBM) and summarize therapeutic targets and approaches that are currently under development. This is a very lengthy extensive manuscript that covers current important topics on GBM based on thorough survey of recent literature. It’s structured as: 1. Introduction, 2. Pathogenetic features, 3. Current treatment, and 4. Treatments in development, and 5. Conclusions. The manuscript is clearly written and equipped with 3 figures and 3 tables that help the reader understand the main messages. Despite these strengths, I find some weaknesses and questions that may warrant attention.

Major points

1. I understand that the authors’ goal was to provide a comprehensive review on GBM without focusing on particular areas, and as a result the manuscript is already very lengthy and the regular reader will find going through difficult. I find this authors’ intention bold and the outcome not necessarily successful. Although the authors seemed to try to cover all current therapeutic strategies that are being tested for GBM, especially clinically, the authors did miss some potentially important ones. For example, agents targeting the DNA damage response such as PARP and ATM inhibitors are essentially ignored, except CDK4/6 inhibitors. I suggest the authors to refer to the recently published review (PMID: 32328653) for a comprehensive summary of key clinical trials in GBM. I think the authors can, or probably should, select therapeutic approaches to discuss, but have to make it clear what are the focuses of this manuscript and what will be not discussed.

2. The other problem arising from the authors’ intention to cover everything is superficial description seen, for example, in PDT and EV sections. The authors have to consider if all the subsections are necessary and useful. In my opinion, comprehensive review on the cutting edge knowledge of GBM takes a book, if the areas covered are to be written with sufficient clarity.

3. I doubt it appropriate to include HDAC inhibitors in agents targeting TP53 pathway. As epigenetic agents, their actions are much more diverse, and certainly not limited to impacts on the TP53 pathways.

Minor points.

4. In Introduction and Conclusions, the authors said that GBM is prevalent across all ages. This is misleading and needs correction. Although it is true that GBM can occur at any age, the incidence of GBM increases with age peaking in the 70’s (see CBTRUS report). In addition, it is clear that the authors’ focus is on adult GBM, since pediatric GBM, which is different from adult GBM, is not discussed at all. This needs clarification in the text.

5. Lines 99-101. The description of GSC differentiation needs accuracy. Do they really differentiate into microglia? Oligodendrocytes may rather be oligodendrocytic since GSCs will not become normal oligodendrocytes.

6. Line 263. Ligand should be receptor?

7. Line 273. TGFbeta is transforming growth factor beta.

8. Line 400. CCNU is not an HDAC inhibitor. What is L01XX38?

9. Line 413. “CDK inhibitors phosphorylate Rb1” is incorrect.

10. Line 438. Is it correct that afatinib is a poor penetrant of BBB?

11. Lines 446-447. Antibody-drug conjugate depatuxizumab mafodotin (ABT-414) is probably worth mentioning.

12. Line 485. The gene name of MET is not considered an abbreviation of mesenchymal epithelial transition, currently. 

13. Line 513. “Interfering interaction with its ligand with ligand-binding icrucumab” is not clear. Please rephrase.

14. Lines 550-551. What is known about the results of clinical trials of proteasome inhibitors?

15. The sentence at lines 556-557 is misleading as this is % of patients with PD-L1+ cells, not % PD-L1 positivity in tumor cells.  

16. Table 3. What is “Anti” next to Atezolizumab?

17. Reference #234 seems an abstract and if so is not appropriate as citation.

18. Lines 646-647. Is “clinical benefits” correct? The references #240, 242, and 243 seem preclinical studies.

19. The paragraph of DC vaccine. The authors should indicate what was loaded to DCs as antigens.

20. Personalized neoantigen vaccine (PMID: 30568305) is worth mentioning.

21. Line 835. What are vessel co-option inhibitors, to be exact?

22. “Effective early diagnostics” is out of the scope of this work, and should be deleted.

Author Response

We sincerely thank the reviewers for evaluating this manuscript. Please find below point-by-point responses to each comment. Editorial formatting may change the line numbers (from what we indicated below) in the word version of the revised manuscript, thus, we request the reviewers to check the PDF version of the revised manuscript.

REVIEWER 3 EVALUATION:

The manuscript by Nguyen et al is a review article that aims to overview pathological and genetic features of glioblastoma (GBM) and summarize therapeutic targets and approaches that are currently under development. This is a very lengthy extensive manuscript that covers current important topics on GBM based on thorough survey of recent literature. It’s structured as: 1. Introduction, 2. Pathogenetic features, 3. Current treatment, and 4. Treatments in development, and 5. Conclusions. The manuscript is clearly written and equipped with 3 figures and 3 tables that help the reader understand the main messages.

Response: We thank the reviewer for highlighting the strengths of this manuscript and finding the review helpful for the readers to understand the main messages.

Despite these strengths, I find some weaknesses and questions that may warrant attention.

Response: Please find below point-by-point responses to the reviewer’s specific comments.  

Major points

  1. I understand that the authors’ goal was to provide a comprehensive review on GBM without focusing on particular areas, and as a result the manuscript is already very lengthy and the regular reader will find going through difficult. I find this authors’ intention bold and the outcome not necessarily successful.

Response: We are thankful to Reviewer 3 for constructive and balanced criticisms. Yes, our intention was bold, but we respectfully disagree that a “regular” reader will find the content difficult. Reviewer 3 indicated earlier that, “the manuscript is clearly written and equipped … [for the] reader [to] understand the main messages.” Based on these comments (plus the statement from Reviewer 1 detailing the manuscript’s clarity and readability), we believe that our paper would be understandable to non-expert readers. 

We have also attempted to provide a comprehensive overview covering all important aspects of GBM’s pathogenetic features and current management, and, thus, it was not possible for us to dive deeply into each and every particular area, which would take a whole book as stated by Reviewers 3 and 4.

Although the authors seemed to try to cover all current therapeutic strategies that are being tested for GBM, especially clinically, the authors did miss some potentially important ones. For example, agents targeting the DNA damage response such as PARP and ATM inhibitors are essentially ignored, except CDK4/6 inhibitors.

Response: We appreciate the reviewer for this note. We have now included a new section briefly describing PARP and ATM inhibitors (lines 629-639), and, accordingly, we have listed these inhibitors in Table 2.

I suggest the authors to refer to the recently published review (PMID: 32328653) for a comprehensive summary of key clinical trials in GBM.

Response: We appreciate the reviewer for bringing this review to our attention. We have added several studies from this article in Table 2 (please see track changes) and cited this paper in the text (line 469).

I think the authors can, or probably should, select therapeutic approaches to discuss, but have to make it clear what are the focuses of this manuscript and what will be not discussed.

Response: We apologize for not understanding the full extent of the reviewer’s comment, “select therapeutic approaches to discuss”.  In this manuscript, in addition to GBM’s pathogenetic features, we have outlined the current status of novel and clinically-relevant therapeutic approaches for GBM such as targeted therapies, various forms of immunotherapies (such as checkpoint inhibitors, oncolytic viruses, therapeutic vaccines, etc.), and a number of other emerging treatment strategies (such as PDT, nanomedicine, EV, miRNA, vessel co-option and vascular mimicry, etc). We have briefly indicated this statement at the end of the Introduction section. We have also now highlighted in the Abstract/Introduction that this manuscript focuses on “adult” GBM.

  1. The other problem arising from the authors’ intention to cover everything is superficial description seen, for example, in PDT and EV sections. The authors have to consider if all the subsections are necessary and useful. In my opinion, comprehensive review on the cutting-edge knowledge of GBM takes a book, if the areas covered are to be written with sufficient clarity.

Response: PDT and EV are emerging therapies in GBM, and there are several clinical studies that have already been conducted to explore their mechanisms and anti-GBM effects. With due respect to those scientists, we would like to keep these two sections as such. However, we want to emphasize that we completely agree with your opinion that it would take a book to cover the cutting-edge knowledge of GBM, and we apologize for not being able to go to the level of depth required based on space constraints.

  1. I doubt it appropriate to include HDAC inhibitors in agents targeting TP53 pathway. As epigenetic agents, their actions are much more diverse, and certainly not limited to impacts on the TP53 pathways.

Response: Thank you for this criticism. We respect the reviewer’s doubts and agree that HDAC inhibitors have a diverse range of action that is not just limited to the TP53 pathway. In this manuscript, we categorized targeted therapies based on their zone of action in 3 dysregulated signaling pathways (p53, RB and RTK). We intended to align these pathways with GBM’s pathogenetic features (i.e., molecular heterogeneity in section 2.2 [3rd paragraph] in the text), and since HDAC inhibitors were shown to destabilize mutant p53 (Ref. # 83 - PMID: 30200436), we included HDAC inhibitors under the p53 pathway.

Minor points.

  1. In Introduction and Conclusions, the authors said that GBM is prevalent across all ages. This is misleading and needs correction. Although it is true that GBM can occur at any age, the incidence of GBM increases with age peaking in the 70’s (see CBTRUS report).

Response: We apologize for this confusion. We have changed our statement in the Introduction (please see lines 37-38) and removed, “that affects patients of all ages”, from the Conclusions.

In addition, it is clear that the authors’ focus is on adult GBM, since pediatric GBM, which is different from adult GBM, is not discussed at all. This needs clarification in the text.

Response: Please see our response above. Our focus is on adult GBM, which we have now stated in both the Introduction and Abstract sections (lines 27 and 74).

  1. Lines 99-101. The description of GSC differentiation needs accuracy. Do they really differentiate into microglia? Oligodendrocytes may rather be oligodendrocytic since GSCs will not become normal oligodendrocytes.

Response: Now lines 106-107. We apologize for the typos. As GSCs express markers for microglia and can differentiate into oligodendrocyte-like cells, they should have been written as microglia-like cells or oligodendrocyte-like cells. We have changed our descriptions and added supportive references.

  1. Line 263. Ligand should be receptor?

Response: Thank you for this correction. The word “ligand” has been replaced by “receptor” in line 301.

  1. Line 273. TGFbeta is transforming growth factor beta.

Response: Thank you for the correction. We have made the appropriate correction (now in line 123).

  1. Line 400. CCNU is not an HDAC inhibitor. What is L01XX38?

Response: Now line 453. Thank you for this note. CCNU is replaced by the HDAC inhibitor ‘trichostatin A’. L01XX38 is also called Vorinostat (an HDAC inhibitor).

  1. Line 413. “CDK inhibitors phosphorylate Rb1” is incorrect.

Response: We apologize for creating this confusion. The statement in line 466 has been changed to, “CDK inhibitor such as palbociclib directly suppresses phosphorylated Rb1.”

  1. Line 438. Is it correct that afatinib is a poor penetrant of BBB?

Response: Now line 502. Yes, it is. Please refer to Ref. # 162 (PMID: 30705084), which we have added to the text.

  1. Lines 446-447. Antibody-drug conjugate depatuxizumab mafodotin (ABT-414) is probably worth mentioning.

Response: Now line 511. We have added ABT-414 in the text with a reference.

  1. Line 485. The gene name of MET is not considered an abbreviation of mesenchymal epithelial transition, currently.

Response: We thank the reviewer for this important update. We changed MET to hepatocyte growth factor receptor (HGFR/c-MET) in line 552.

  1. Line 513. “Interfering interaction with its ligand with ligand-binding icrucumab” is not clear. Please rephrase.

Response: Now line 582. We have rephrased this statement.

  1. Lines 550-551. What is known about the results of clinical trials of proteasome inhibitors?

Response: We have added one more study about proteasome inhibitors to lines 626-627.

  1. The sentence at lines 556-557 is misleading as this is % of patients with PD-L1+ cells, not % PD-L1 positivity in tumor cells.  

Response: We apologize for the confusion. We have modified our statement in lines 644-645.

  1. Table 3. What is “Anti” next to Atezolizumab?

Response: It should actually be “anti-PD-L1”. The word “PD-L1” may have been missed during editorial formatting of the table.

  1. Reference #234 seems an abstract and if so is not appropriate as citation.

Response: Now it is Ref. # 245. We are not aware of any publication by this group presenting the Phase II study results other than in abstract form.

  1. Lines 646-647. Is “clinical benefits” correct? The references #240, 242, and 243 seem preclinical studies.

Response: Now line 782. We appreciate this note and apologize for the typo. We have changed our statement to “pre-clinical”.

  1. The paragraph of DC vaccine. The authors should indicate what was loaded to DCs as antigens.

Response: Now lines 789-790. We have listed antigens that can be loaded in DCs.

  1. Personalized neoantigen vaccine (PMID: 30568305) is worth mentioning.

Response: We have added this study in lines 832-835.

  1. Line 835. What are vessel co-option inhibitors, to be exact?

Response: This study inhibited vessel co-option by utilizing a porcupine inhibitor LGK974, which blocks Wnt secretion. We have added this drug in the text in lines 990-991.

  1. “Effective early diagnostics” is out of the scope of this work, and should be deleted.

Response: We have deleted this statement.

Reviewer 4 Report

This is a quite successful attempt to summarize the recent literature to understand the heterogeneous features of GBM and their potential role in treatment resistance.

The undertaking is almost impossible, but the authors still did a good job!

I think that for those who are a true expert in GBM treatments it may seem a reductive work, but for those who do not have good knowledge in this topic it can be an excellent starting point.

As mentioned above, there are no comprehensive reviews of this type, but they are focused on a specific aspect. This work, on the other hand, is all-encompassing on current therapies and also deals with therapies under current development (the main ones!). Maybe he doesn't go into all of them in depth, but the review is already very broad so ... it would take a whole book to explore the whole topic in depth. However, the only drawback would be to specify better that the review is focused on adult cancers and not children. And I recommend inserting a list with the abbreviations used perhaps at the beginning of the article

I think the paper is well written, very clear and therefore accessible to a wider range of readers such as medical students.
Finally, the conclusions are consistent and the question posed have been argued.

Author Response

We sincerely thank the reviewers for evaluating this manuscript. Please find below point-by-point responses to each comment. Editorial formatting may change the line numbers (from what we indicated below) in the word version of the revised manuscript, thus, we request the reviewers to check the PDF version of the revised manuscript.

REVIEWER 4 EVALUATION:

This is a quite successful attempt to summarize the recent literature to understand the heterogeneous features of GBM and their potential role in treatment resistance.

Response: We thank the reviewer for this positive comment.

The undertaking is almost impossible, but the authors still did a good job!

Response: We appreciate the reviewer for a balanced/mixed criticism, but we are confused with “almost impossible statement” based on Reviewer 4’s last statement, “I think the paper is well written, very clear and therefore accessible to a wider range of readers.” As stated earlier in response to Reviewer # 3, we have provided a comprehensive overview covering all important aspects of GBM’s pathogenetic features and current management, and, thus, it was not possible for us to deeply explore every particular area, which would take a whole book as also confirmed by Reviewer # 3.

I think that for those who are a true expert in GBM treatments it may seem a reductive work, but for those who do not have good knowledge in this topic it can be an excellent starting point.

Response: We respectfully disagree with the “reductive work” statement because we feel that we did cover the basic concepts of GBM pathogenetic features (although briefly due to space limitations), along with various mechanisms of action of different therapeutic approaches that are currently being used in the field. As Reviewer 4 indicated below, “the paper can also be beneficial to a wide range of reader”, and we totally agree with this specific statement.

As mentioned above, there are no comprehensive reviews of this type, but they are focused on a specific aspect. This work, on the other hand, is all-encompassing on current therapies and also deals with therapies under current development (the main ones!).

Response: We very much appreciate the reviewer for this positive feedback.

Maybe he doesn't go into all of them in depth, but the review is already very broad so ... it would take a whole book to explore the whole topic in depth.

Response: We completely agree with this statement. Please see our responses above.

However, the only drawback would be to specify better that the review is focused on adult cancers and not children.

Response: This (adult GBM) has now been specified in the Abstract and Introduction sections (lines 27 and 74).

And I recommend inserting a list with the abbreviations used perhaps at the beginning of the article

Response: In this manuscript, we indicated the abbreviations whenever they were mentioned for the first time. However, we agree with the reviewer that it would be easier for the readers if all abbreviations are added at the very beginning of the manuscript text.  We are happy to add all abbreviations at the beginning, but we leave this decision up to the Editor.

I think the paper is well written, very clear and therefore accessible to a wider range of readers such as medical students.
Finally, the conclusions are consistent and the question posed have been argued.

Response: Thank you for this positive feedback.

Round 2

Reviewer 3 Report

  1. Line 102. I found reference #35 is pretty weak evidence for microglia-like differentiation of GSCs as they only looked at expression of SPARC as a microglial maker, which is probably not appropriate given well documented role of SPARC in GBM invasion. There exists more solid literature for GSC differentiation towards neuronal, endothelial and pericytic cells.
  2. Line 404. Replace L01XX38 with vorinostat or indicate vorinostat in parenthesis so that the term is consistent with Table 3.
  3. Lines 573-574. "with PD-L1 expression in tumor cells is heterogenous and varies from 61 to 88% of GBM patients" is not grammatically right. Please correct.  

Author Response

We sincerely thank Reviewer 3 for critically reviewing the revised manuscript. Changes are highlighted in yellow in the text.

Response to Reviewer 3:

 Line 102. I found reference #35 is pretty weak evidence for microglia-like differentiation of GSCs as they only looked at expression of SPARC as a microglial maker, which is probably not appropriate given well documented role of SPARC in GBM invasion. There exists more solid literature for GSC differentiation towards neuronal, endothelial and pericytic cells.

 Response: Thank you for your comment. We have added four relevant references (please see below) demonstrating GSC differentiation towards neuronal, endothelial and pericyte cells.

  1. Cheng, L.; Huang, Z.; Zhou, W.; Wu, Q.; Donnola, S.; Liu, J.K.; Fang, X.; Sloan, A.E.; Mao, Y.; Lathia, J.D., et al. Glioblastoma stem cells generate vascular pericytes to support vessel function and tumor growth. Cell 2013, 153, 139-152, doi:10.1016/j.cell.2013.02.021.
  2. Xu, J.; Gong, T.; Heng, B.C.; Zhang, C.F. A systematic review: differentiation of stem cells into functional pericytes. The FASEB Journal 2017, 31, 1775-1786, doi:https://doi.org/10.1096/fj.201600951RRR.
  3. Mei, X.; Chen, Y.-S.; Chen, F.-R.; Xi, S.-Y.; Chen, Z.-P. Glioblastoma stem cell differentiation into endothelial cells evidenced through live-cell imaging. Neuro-Oncology 2017, 19, 1109-1118, doi:10.1093/neuonc/nox016.
  4. Beier, C.P.; Rasmussen, T.; Dahlrot, R.H.; Tenstad, H.B.; Aarø, J.S.; Sørensen, M.F.; Heimisdóttir, S.B.; Sørensen, M.D.; Svenningsen, P.; Riemenschneider, M.J., et al. Aberrant neuronal differentiation is common in glioma but is associated neither with epileptic seizures nor with better survival. Scientific Reports 2018, 8, 14965, doi:10.1038/s41598-018-33282-5.

Line 404. Replace L01XX38 with vorinostat or indicate vorinostat in parenthesis so that the term is consistent with Table 3.

Response: Replaced L01XX38 with vorinostat. Thank you.

Lines 573-574. "with PD-L1 expression in tumor cells is heterogenous and varies from 61 to 88% of GBM patients" is not grammatically right. Please correct.  

Response: The sentence has been corrected to “with varying degree of PD-L1 expression in tumor cells in GBM patients ranging from 61% to 88%.” Thank you.